# Biological and environmental rhythms in (dark) deep-sea hydrothermal ecosystems

Daphne Cuvelier[1,2]*, Pierre Legendre[3], Agathe Laes-Huon[4], Pierre-Marie Sarradin[1], Jozée Sarrazin[1]

[1.] Ifremer, Centre de Bretagne, REM/EEP, Laboratoire Environnement Profond, Plouzané, 29280, France
[2.] Mare - Marine and Environmental Sciences Centre, Department of Oceanography and Fisheries, Rua Professor Frederico Machado 4, Horta, 9901-862, Portugal §
[3.] Département de Sciences Biologiques, Université de Montréal, C.P. 6128, succursale Centre-ville, Montréal, H3C 3J7, Québec, Canada
[4.] Ifremer, Centre de Bretagne, REM/RDT, Laboratoire Détection, Capteurs et Mesures, Plouzané, 29280, France

*correspondence to: Daphne Cuvelier (daphne.cuvelier@gmail.com)
§ Current address of corresponding author

**Abstract**

During 2011, two deep-sea observatories focusing on hydrothermal vent ecology were up and running in the Atlantic (Eiffel Tower, Lucky Strike vent field) and the North-East Pacific Ocean (NEP) (Grotto, Main Endeavour field). Both ecological modules recorded imagery and environmental variables jointly for a time span of 23 days (7-30 October 2011) and environmental variables for up to 9 months (October 2011 - June 2012). Community dynamics were assessed based on imagery analysis and rhythms in temporal variation for both fauna and environment were revealed. Tidal rhythms were found to be at play in the two settings and were most visible in temperature and tubeworm appearances (at NEP). A ~6-hour lag in tidal rhythm occurrence was observed between Pacific and Atlantic hydrothermal vents which corresponds to the geographical distance and time delay between the two sites.

## 1. Introduction

All over our planet, animals are influenced by day- and night-cycles. Entrainment occurs when rhythmic physiological or behavioural events in animals match the periods and phase of an environmental oscillation, e.g. circadian rhythms to light-dark cycles. In marine populations such cycles are evident in the photic zone (Naylor 1985). However, more recently similar cycles have become apparent in deep-sea organisms and populations as well, at depths where light does not penetrate. At these greater depths, fluctuations in light intensity are likely to be replaced by changes in hydrodynamic conditions (Aguzzi et al., 2010). Several studies reveal the presence of tidal cycles in environmental variables (such as currents, fluid emission, temperature) in the deep sea, particularly at hydrothermal vents (e.g. Tivey et al., 2002; Thomsen et al., 2012; Barreyre et al., 2014; Sarrazin et al., 2014; Lelièvre et al., 2017) and the influence of tides on the deep-sea organisms has been previously inferred. In meantime, an actual tidal rhythm has been revealed in visible faunal densities and appearance rate for inhabitants of various deep-sea chemosynthetic environments (e.g. a semi-diurnal tidal component in buccinids at cold seeps (Aguzzi et al., 2010) and semi-diurnal and diurnal tidal components in siboglinids (Tunnicliffe et al., 1990;

Cuvelier et al., 2014)). Presumably, though difficult to statistically demonstrate, the deep-sea organisms respond
to or reflect the changing surrounding environmental conditions, which are modulated by hydrodynamic processes
including the tides.
Despite the growing realisation that tidal influences are indeed at play in the deep ocean, it remains hard to actually
reveal these patterns because of the isolation of the ecosystem and the limited access to the longer time-series. The
use of deep-sea observatories, which have been deployed recently in various seas and oceans (see Puillat et al.,
2012 for an overview) brings out new insights into the dynamics of these remote habitats. First ecological analyses
based on deep-sea observatories have been published (Juniper et al., 2013; Matabos et al., 2014; 2015; Cuvelier et
al., 2014; Sarrazin et al., 2014; Lelièvre et al., 2017), and many more works are in progress.
The current observatory-based study allows a unique comparison of hydrothermal vent community dynamics
between two different oceans featuring a different seafloor spreading rate. Data originating from the deep-sea
observatories on the slow-spreading Mid-Atlantic Ridge (MoMAR, now EMSO-Açores) and on the faster-
spreading Juan de Fuca Ridge (North-East Pacific, NEPTUNE, now called Ocean Networks Canada (ONC)),
featuring the same time span and resolution, have been analysed. The two oceans are characterised by different
vent fauna, with a visual predominance of *Bathymodiolus* mussel in the shallower (<2300m) Atlantic and *Ridgeia*
tubeworms in the North-East Pacific, but they do share higher taxonomic groups. Following key questions are put
forward: (i) Are there rhythms discernible in both hydrothermal settings? (ii) Is there a lag/time difference in
community dynamics and environmental variables observed between the two oceans? (iii) Which environmental
variables influence community dynamics? and finally (iv) Do the shared taxa occupy similar microhabitats and
possible niches in each ocean? Answering these questions will provide new insights in understanding local vent
community dynamics and will enlighten us on similarities and differences between oceanic ridges and oceans. In
order to do this, a dual approach was wielded, assessing a short-term comparison between fauna and environment
(23 days) and a longer-term comparison of environmental variables (9 months) featuring the same observation
window at both study sites.

**2. Material and Methods**
**2.1. Observatories and study sites**
Two similar ecological observatory modules, called TEMPO and TEMPO-mini were deployed in two different
oceans in 2011 (Fig. 1). The first one (TEMPO) was part of the EMSO-Azores observatory (http://www.emso-
fr.org/EMSO-Azores) and was deployed on the Lucky Strike vent field on the Mid-Atlantic (MAR) Ridge, south
of the Azores. The wireless EMSO-Azores observatory consists of two main hubs, positioned east and west of the
central lava lake that is characteristic of the Lucky Strike vent field. The eastern hub (Seamon East, Blandin et al.,
2010) focuses on hydrothermal vent ecology and hosts the TEMPO module. TEMPO 2011 was positioned at
1694m depth at the southern base of a large 11m high hydrothermally active edifice called Eiffel Tower. Its
counterpart, TEMPO-mini, was implemented on the region-scaled cabled network NEPTUNE
(http://www.oceannetworks.ca/) in the North-East Pacific (NEP), as part of the Endeavour instrument node. It was
deployed at a depth of 2168m on a small 5m high platform on the north slope of the Grotto hydrothermal vent, a
10m high active edifice at Main Endeavour Field (MEF). Both modules were equipped with a video camera (Axis
Q1755), temperature probes, a CHEMINI Fe analyser (Vuillemin et al., 2009) and an optode measuring
temperature and oxygen. An additional instrument measuring turbidity was deployed in the vicinity of the TEMPO
module in 2011 (Table 1). The biggest discrepancy between both modules was the energy provision, with the
Atlantic one (TEMPO) being autonomous and battery-dependent (wireless), and the North-East Pacific one
(TEMPO-mini) being connected to a cabled network. Detailed descriptions of both modules can be found in
Sarrazin et al. (2007, 2014) for TEMPO and Auffret et al. (2009) and Cuvelier et al. (2014) for TEMPO-mini.
Henceforth, the Atlantic set-up (TEMPO on MoMAR/EMSO-Azores) will be referred to as MAR, and the North-
East Pacific (TEMPO-mini on NEPTUNE/ONC) set-up as NEP (Fig. 1).

**2.2. Data collection and recordings**
Data collected consisted of video imagery recordings, temperature measurements, iron and oxygen concentrations,
and turbidity measurements (the latter for MAR only) (Table 1), which were recorded jointly for the period 7-30
October 2011. Differences in recording resolutions were mainly due to different observatory set-ups and more
particularly due to the cabled or wireless network characteristics and their inherent energy limitations (continuous
power vs. battery dependence). Lights were powered on with the same frequency as the imagery recording (every
6h) at MAR, contrastingly to NEP where lights were on continuously during the period analysed (23 days). At
NEP, TEMPO-mini was equipped with a thermistor array of which two probes (T602 and T603) were deployed
on an assemblage most similar to the one filmed (see Cuvelier et al., 2014). Therefore, only those two probes were
used in the comparison to the MAR temperature data, which was recorded directly on the filmed assemblage.

Iron (from here on referred to as Fe) concentrations were measured on top of the assemblage and within the field
of view (FOV) at MAR (Laës-Huon et al., 2015; Sarradin et al., 2015) and below the FOV at NEP. An *in situ*
calibration was performed at NEP, analysing 2 Fe standards a day of 20 and 60 µmol/l; no such calibration took
place at MAR. At NEP, sampling frequency was changed from twice (30 September - 18 October 2011) to once a
day (19 October 2011 - 31 January 2012) due to rapidly decreasing reagents. Fe concentrations were analysed for
the longer-term and used to explore the differences between the observatory settings.

Closer examination of data recorded by the optode revealed some inconsistencies between the measured
temperature and the $O_2$ concentrations. As the $O_2$ concentrations were corrected by the temperature, a difference
in the response time between the temperature and oxygen sensor within the same instrument was presumed. This
lag could not be quantified, making comparisons with other observations impossible. Oxygen concentrations
measured were thus merely used as illustration to compare the differences between the two hydrothermal settings.

Turbidity was only measured at the MAR observatory in Nephelometric Turbidity Units (NTU), which were
straightforward in their interpretation, i.e. the higher the more turbid. The sensor was not calibrated as such since
its response depended on the particle size, which was unknown. Hence it only provided information on the relative
turbidity (and peaks) of the environment.

## 2.3. Short-term temporal analyses

A unique subset of comparable data, allowing a joint assessment of fauna and environment, was available for the time period 7-30 October 2011 for both observatories. The image analysis period was limited because of data availability, which in this case was restricted by the imagery recordings from NEP (see Cuvelier et al., 2014).

124

### 2.3.1. Imagery analysis

The variations occurring in the faunal assemblages in the two hydrothermal vent settings were analysed for 23 days. For this period, a screen still was taken every 6 hours at 00.00, 06.00, 12.00, 18.00 UTC. For each site, these screen-stills were used as a template in Photoshop© to map and count faunal abundances. Faunal densities were quantified at a 6h frequency, while the microbial coverage was assessed every 12h. To pursue the latter, the microbial cover was marked in white and the rest of the image rendered in black. Using the "magic wand tool" of the ImageJ image analysis software (Rasband, 2012), the surface covered by microorganisms was quantified and converted to percentages. Due to gaps in the data recordings different numbers of images were analysed for MAR and NEP (Table 2). These gaps were failed recordings (due to observatory black-out or instrument failure) or unusable video sequences (empty files, black or unfocused videos).

135

The surface filmed by each observatory was different (Table 2), which is why densities (individuals/m$^2$) were used instead of abundances. In each setting, there was also a discrepancy between the surface filmed and that analysed (Table 2, Fig. 2). Some surfaces were not taken into account because of their increased distance to the camera, the focal point and associated light emission (referred to as 'background'), or due to the probe positioning within the FOV, making it impossible to quantify the fauna. These surfaces were marked in black and white on the map in Fig. 2 and were not included in the analysed surface calculations. Both maps were made based on a composed image, i.e. a merge of all images analysed, hence showing the most recurrent species distributions. For MAR, main shrimp cluster/distribution was confirmed using Matabos et al. (2015). For NEP, heat maps from Cuvelier et al. (2014) were used to confirm and localise mobile fauna. This did not mean that the mobile fauna did not venture elsewhere, but it showed an average distribution.

146

The Atlantic and Pacific oceans feature distinct hydrothermal vent fauna and while they do share several higher-level taxa, most species are different for the two oceans (Fig. 1 and 2). The main visible species and engineering taxon present for the 'shallower' (<2300m) Mid-Alantic vents is a mytilid (*Bathymodiolus azoricus*) versus a siboglinid tubeworm for the NEP (*Ridgeia piscesae*). The second most characteristic Atlantic taxon is the *Mirocaris fortunata* alvinocaridid shrimp (Desbruyères et al., 2001; Cuvelier et al., 2009). Contrastingly, no hydrothermal shrimp are present at NEP vents, but associated visible fauna consisted of Buccinidae (Gastropoda), Polynoidae (Polychaeta) and Pycnogonida (containing the family Ammotheidae) (Cuvelier et al., 2014; Table 2, Fig. 2). The latter two taxa are also present at the shallower MAR sites be it in lower abundances and represented by different genera and species, as well as a bucciniform gastropod (Turridae family). In the Atlantic FOV, a small patch of anemones (Actiniaria) was visible below the probe as well as single occurrences of Ophiuroidea. Visiting

fish species consisted of *Cateatyx laticeps* (Bythitidae) and *Pachycara* sp. (Zoarcidae) at MAR and NEP respectively. *Segonzacia mesatlantica* (Bythograeidae) crabs were abundant at MAR while Majid spider crabs could be occasionally observed at NEP.

Overall, imagery analysis was limited to the density assessment of the visible species (Cuvelier et al., 2012). In this perspective, tubeworm densities corresponded to the number of visible tubeworms, i.e. those that had their branchial plumes out of their tube at the moment of the image analysis. From here on, tubeworms visibly outside of their tubes will be referred to as tubeworm densities. Stacked limpets were visible on the NEP imagery but were impossible to assess quantitatively due to their small size and piling (Cuvelier et al., 2014).

### 2.3.2. Environmental data

An active fluid exit was visible on the images of the MAR, but not on the NEP recordings. The probe measuring the MAR environmental variables was positioned next to this fluid exit in the FOV, whilst the different probes of NEP (multiple probes measuring different environmental variables, see *in situ* observatory set-up in Cuvelier et al., 2014) were deployed below the FOV. The frequencies with which the environmental variables were recorded were listed in Table 1. Due to the large variability and steep gradients in environmental conditions observed in the hydrothermal vent ecosystems, the temperature variables used in the analyses were averaged per hour to reduce noise and variance. Only probes T602-T603 from NEP were used for comparison with MAR. The R package hydroTSM (Zambrano-Bigiarini, 2012) was used to create an overview of the variations of hourly temperature values during imagery duration. For those variables used as explanatory variable (temperature and turbidity) in the joint analyses with the available faunal densities, every $6^{th}$h value was taken (corresponding with the 6h frequency at 00.00, 06.00, 12.00 and 18.00 UTC). Fe was only sampled with a 12h or 24h frequency, hence limiting its use as an explanatory variable for the higher resolution faunal dynamics.

### 2.3.3. Statistical analyses

Multivariate regression trees (MRT, De'ath, 2002) were computed on Hellinger-transformed faunal densities. This analysis is a partitioning method of the species density matrix of each observatory, constrained by time. It grouped consistent temporal observations and thus identified groups with similar faunal composition that were adjacent in time; these groups were called "temporal split groups" from here on. Each split was chosen to maximise the among-group sum-of-squares and the number of split groups was decided upon by choosing the tree with the lowest cross-validation error; that tree had the best predictive power. For this type of analysis, the observations did not need to be equi-spaced, as long as the constraining variable reflects the sampling time (Legendre and Legendre, 2012). The MRT partition was then subjected to a search for indicator taxa (IndVal analysis, Dufrêne and Legendre, 1997; function multipatt in R package Indicspecies (De Caceres and Legendre, 2009)). The IndVal index combined a measure of taxon specificity with a measure of fidelity to a group and thus revealed which taxon was significantly more or less abundant in the group before than after the split. Its significance was assessed *a posteriori* through a permutation test (Borcard et al., 2011). The observed temporally consistent groups were delineated by colour-codes within a Redundancy Analysis (RDA) ordination plot; RDA's were carried out on the Hellinger transformed faunal densities and environmental variables to visualise the possible influence of the environmental constraints

on the temporal groups found in the faunal density matrices. Environmental variables were subject to forward
selection (packfor package in R, Dray, 2009), revealing those explaining most of the variation in faunal
densities ($\alpha$=5%).
Rhythms and periodicities in faunal densities and environmental variables were examined with Whittaker-
Robinson (WR) periodograms (Legendre, 2012). These WR periodograms were computed on the faunal densities,
with a 6h resolution, and on the environmental variables with an hourly resolution (see 2.4). Prior to these analyses,
stationarity was implemented by detrending time series when necessary. Time series were folded into Buys-Ballot
tables with periods of 2 to a maximum of n/2 observations. The WR amplitude statistic was the standard deviation
of the means of the columns of the Buys-Ballot table. Missing values were taken into account and filled in by NA
values (''Not Available'').
In order to establish differences or similarities in the variations observed in temperature data from MAR and NEP,
cross-correlations were carried out on the hourly temperature data for imagery duration (n=553). Cross-
correlations could not be carried out between faunal and environmental variables, because the time series were
relatively short and they contained gaps, an irregularity which cross-correlations cannot take into account.
No specific correlations between faunal densities and environmental variables were presented. The high spatial
variation occurring at hydrothermal vents proved difficult to capture with the experimental settings from the 2011
deployments. The probes at NEP were placed at a distance from the filmed assemblage and the relatively large
surface filmed at MAR decreased the representativeness of single point measurements. The measurements made
were considered more representative of an overall variability but not necessarily at the scale of individuals.
Structuring strength and tendencies of environmental variables in faunal composition were deduced from
ordinations.
**2.4. Long-term temporal analyses**
For the time period 29 September 2011 to 19 June 2012, environmental data spanning 9 months of temperature
and Fe were available for compared analyses, turbidity was only available for the MAR. The oxygen time series
revealed the issues explained previously (see 2.2) and were not subject to temporal analyses but the differences in
concentrations measured between the two observatory locations were addressed. Faunal densities could not be
assessed on the longer term due to the lack of regular imagery recordings for MAR and NEP but also changes in
zoom and subsequently image quality for the NEP. Long-term time series analyses in the form of WR
periodograms were carried out on the hourly data for temperature and turbidity, and daily/12h (NEP/MAR
respectively) frequency for Fe to allow comparison between MAR and NEP. See section 2.3.3 for details on the
periodogram analyses.

## 3. Results

### 3.1. Short-term variability

#### 3.1.1. Fauna

**MAR** – In total, 84 images were analysed from the TEMPO module; there were 9 gaps in the imagery data series (Table 2). The most abundant visible species were *Bathymodiolus azoricus* mussels and *Mirocaris fortunata* shrimp, the numbers of the other taxa (crabs, polynoids, bucciniform gastropods, pycnogonids) being an order of magnitude smaller (hundreds vs. single occurrences, for densities see Fig. 3.). An overall significant increase in mussel and shrimp densities was observed ($R^2$=0.68, p <0.001 and $R^2$=0.32, p <0.001 respectively, Fig. 3). Conversely, a significant negative trend was observed for the bucciniform gastropods ($R^2$=0.19, p-value<0.001, Fig. 3). For the other taxa, no significant trends in densities were observed. Trends were removed prior to periodogram analyses, which revealed no significant rhythms in mussels, shrimp, crabs and bucciniform gastropods. Only for polynoid scale worms, a significant 18h period was observed, followed by significant periods at 90h (5x18h), 186 h (~10x18h) and 204h (~11x18h) (Fig. S1). Polynoids were mostly found on bare substratum though they ventured on the mussel bed occasionally. In fact, 92% of the observations were associated with bare substratum vs. 8% observations on the mussel bed. One large individual occupied the exact same area in 61% of all images analysed (Fig. 2). Bucciniform gastropods were observed on the bare rock in the foreground further away from the fluid exit (Fig. 2). Pycnogonids (7 observations) and the occasional ophiuroid (4 observations) were observed mostly at the edge or on top of the mussel bed, further away from fluid flow. *Segonzacia mesatlantica* crabs were mobile, some moving in the FOV, others appearing between the mussels. Their distribution was rather heterogeneous but mostly associated with the mussel beds and shrimp presence. A *Cataetyx laticeps* fish was observed 5 times within the analysed time series – mostly in the background and not interacting actively with the other organisms. Its presence was only discernible based on the video footage (and not on the screen stills). The small patch of anemones observed below the probe featured 33 individuals. No changes were documented over time for this taxon.

**NEP** – 88 images were analysed from the TEMPO-mini module; there were 5 gaps in the imagery dataset (Table 2). *Ridgeia piscesae* tubeworms were the most abundant taxon assessed on imagery, adding up to several hundred visible (outside their tubes) individuals and with their tubes providing a secondary surface for the other organisms to occupy. Thus, several dozens of pycnogonids, up to a dozen of polynoids and a couple of buccinids were present on the tubeworm bush (for densities see Fig. 3.). The strings of stacked limpets were not quantified. Only pycnogonid densities showed a significant positive temporal trend ($R^2$=0.23, p <0.001, Fig. 3). For the other taxa, no significant trends were observed. Periodogram analyses carried out on the faunal densities with a 6h period revealed a distinct 12h frequency and harmonics for tubeworms, a single 12h period (i.e. no harmonics) and 222h (9.25 days) for polynoids (Fig. S2). Buccinids also showed some significant frequencies at 174h (7.25 days) and 204-228h (~8.8 days, Fig. S2), while none were observed for pycnogonids. Pycnogonids showed distinct clustering behaviour and spatial segregation which were also observed for the other taxa (buccinids and polynoids), be it to a lesser extent. Eight visits of a *Pachycara* sp. (Zoarcidae) were documented, during which the fish was present next to the tubeworm bush and sometimes hiding underneath it. No specific behaviour of the fish interfering with the fauna of the tubeworm bush was documented.


**Temporal split groups** - Different adjacent temporal groups were identified for MAR and NEP based on changes
in faunal composition and densities over time through Multivariate Regression Trees (MRT). Five temporal groups
were delineated for NEP and MAR (Table 3) though they were partitioned differently over time. Most groups
could be considered rather similar in time span for the two locations. For the MAR, the highest variance was
described by the split separating <195h and ≥195h. This coincided with an increase in shrimp and mussel densities
and decrease in gastropods and crab densities (Fig. 3), which were shown to be significantly indicative for different
split groups post-195h. Shrimp were found to be most indicative for the ≥321h group (IndVal=0.47, $p<0.05$) and
bucciniform gastropods for the ≥195h-<321h group (IndVal =0.78, $p<0.001$). *Bathymodiolus* mussels were
indicative for the <51h group (IndVal =0.45, $p<0.05$) featuring the lowest densities for the studied time series.
Contrastingly for the NEP, splits coincided with the chronology and tubeworm densities were significantly
indicative for the <45h group (IndVal =0.46, $p<0.001$). Pycnogonids and buccinids were both indicative of ≥504h
(IndVal=0.51, $p<0.001$ and IndVal=0.52, $p<0.001$ respectively). The temporal split groups (Table 3) were
delineated onto the faunal variation graphs (Fig. 3) and used to colour-code groups in the ordinations (see 3.1.3)
in order to investigate how individual taxa and environmental conditions coincide with and influence the temporal
inconsistencies represented by the MRT groups.

**Microbial Cover** - Despite the large difference in percentage of the image covered by microbial mats between
MAR (1.34–2.76%) and NEP (25.11–37.02%), both showed a decline during the period analysed (Fig. 4). The
observed trends were significantly negative for both sites. For the MAR, this decline corresponded to a significant
negative correlation between microbial cover and mussel densities ($r=-0.67$, $p<0.001$) and shrimp densities ($r=-$
$0.53$, $p<0.001$). For the NEP, no significant correlations between microbial cover and other taxa were revealed.

**3.1.2. Environmental data**
Environmental data analysis presented in this section is a short-term analysis, spanning 23 days corresponding to
the imagery duration.

**Temperature -** Generally, higher temperatures were recorded at the MAR (Fig. 5). Mean temperatures at MAR
were significantly higher than maxima recorded by probes T602 and T603 at NEP (Fig. 5, Table 4), coinciding
with higher ambient seawater temperatures for the MAR (~4°C) than for NEP (~2°C). Even when rescaling to
ambient temperature, minimum temperatures measured on the MAR were still higher than those of the NEP.
However, maximum and mean temperatures no longer stood out (but remained significantly different at $p<0.05$)
and were even lower than those measured by probes T602 and T603 in the NEP (Table 4). Standard deviations
and variance were maintained and were consistently higher at NEP, but not significantly different.

The hourly temperature recordings showed noticeable cycles of higher and lower temperatures specifically in T602
and T603 (visible as red and blue colours in Fig. 6 respectively). When such (more or less) coherent bands of lower
and higher values are observed in tidal pressure heat-maps, it shows the cyclical nature of the tides. Hence,
alongside the tidal rhythms revealed by the periodogram analyses, a tidal cyclicity was recognisable in the

temperature recordings of the NEP. Patterns were less clear for the MAR temperature data. Information on pressure data from the same localities and correspondence to the temperature measurements was included as appendix/supplementary material (Fig. S3).

In order to investigate how the temperature time series from the two oceans related to one another, cross-correlations were carried out on the hourly temperature values (Fig. 7). Generally, positive autocorrelations were more pronounced, meaning that the two series were in phase. Maximum autocorrelation was reached at lag +5 h when comparing MAR to T602 with the MAR time series leading, and a +5 to +6 h lag between MAR and T603. Most of the dominant cross-correlations occurred between lags +4 and +7, with tapering occurring in both directions from that peak. This corresponded to the time difference of ~6h between MAR and NEP locations, calculated as follows: 24*degrees (difference in longitude)/360. Maximum negative autocorrelations were observed at lags –14 and +11 for NEP T602 and MAR and between lags +10 and +13 for NEP T603 and MAR. The difference between the maxima (and minima) closely corresponded to the tidal cycle (~6h).

**Fe** – There was a 6 hours' time difference in the Fe-recordings carried out in the NEP being measured at 6.00 and 18.00 UTC and on the MAR at 12.00 and 00.00 UTC. Fe on the MAR was recorded twice a day (in 4 cycles) during the analysed imagery period. Concentrations ranged from 0.41 µmol/l to 1.62 µmol/l with a mean of 0.81±0.28 µmol/l. A non-significant ($p>0.4$) positive trend was observed but no significant relationships between fauna, microbial cover and Fe were revealed. Fe measurements at NEP were limited to 7 days at a frequency of one measurement a day (Fig. 5). Consequently, its use as an explanatory variable for faunal variations was limited and no patterns were revealed. Values ranged from 2.07 to 2.99 µmol/l, which were higher than those observed on the MAR but also showed less variation.

**Turbidity** – Turbidity measurements (NTU) were restricted to the MAR observatory and a non-significant positive trend was observed during imagery duration. A large peak was noticeable at ~400 h (around 23 October 2011) though it was not reflected in any of the other environmental variables or community dynamics (Fig. 5).

### 3.1.3. Fauna-environment interaction

**MAR -** Environmental variables incorporated in the ordination analyses did not distinguish significantly between faunal densities or the temporal split groups found in the faunal composition (Fig. 8). The first axis was most important for the MAR RDA (83.76%), hence attributing a higher importance to the horizontal spreading, but was not significant ($p>0.05$). This separation corresponded mostly with the separation of *Mirocaris* and *Bathymodiolus*. NTU seemed to have a distinct impact on separating the images from one temporal split group (from 51h to 195h), though there was no clear signal in NTU values at that time. Overall, for the MAR, no distinct relationship between a specific taxon and measured environmental variables was revealed.

**NEP -** The first axis of the NEP RDA was significant at $p<0.005$ and explained most of the variance (98.2%) represented by the ordination plot (Fig. 8). This coincided with a separation in the plot between Pycnogonida, Polynoidae and Buccinidae that pooled apart from the tubeworms. This lateral separation in taxa coincided with

the strong correlation between tubeworm densities (appearances) and the T602 and T603 temperature measurements. Only T603 was significant at p<0.05. Temporal split groups were vertically aligned in the plot and tended to overlap, with tubeworms being more indicative for <45h group (as corroborated by the "multipatt indval" analysis). No clear influence from the environmental variables on the separation in temporal split groups could be revealed.

**3.2. Long-term variability**

Long-term variations in environmental conditions from both observatories spanning 9 months were investigated. As for the short-term analysis, the long-term time series analysed was limited by the shortest deployment period for which both observatories were up and running at the same time and was thus restricted by the TEMPO-mini observatory (NEP).

**Temperature** - The continuous MAR temperature time series showed temperature variations between 4.48 and 10.91°C, with a mean of 5.54±0.71°C (Fig. 9). A significant negative trend in temperature values was observed over the 9-month period; a trend already visible in the short-term analyses. The NEP temperature values recorded during this period by T602 and T603 were comprised between 2.23°C and 5.43°C, with a mean of 3.78±0.54°C. T602 showed a significant negative trend (p<0.001) while T603 showed a significant positive trend (p<0.001) over the longer term. Trends were removed and periodogram analysis was carried out on the residuals for periods of 2 to n/2 (3168h ~ 4.5 months), 2 to 800h (~1 month), and 1 week periods (2 to 200h). Regardless of the time-span, diurnal and semi-diurnal periods and their harmonics were the main significant frequencies discerned. No clear or distinct significant hebdomadal (weekly) or infradian (multiple days) cycles were encountered. Therefore, in order to facilitate interpretation, only the periodograms with periods of 2 to 200h were presented (Fig. 10).

A significant period at 12h was revealed for the MAR and NEP T602 probes, but not for T603. For T603, a peak was present at T=12h but it was not significant; however, harmonics of that peak at 25, 37, 50 and 74, 75h (etc.) were significant (Fig. 10). A significant 25h period was thus observed for both NEP probes (T602 and T603). Recurrent harmonics of both semi-diurnal (12h) and diurnal (25h) frequencies were identifiable throughout the temperature time series, more so for NEP time series than for MAR, which agree well with the tidal cycle (12h 25 min and 24h 50 min) (Fig. 10). A distinct 6.25-day period (at 150h) with a high amplitude was revealed for the T602 and T603 probes (Fig. 10). Such a peak was recognisable for the MAR as well, though it was not significant. A peak at 174h (7.25 days) was significant for all three probes (MAR and NEP). The corresponding significant periods between MAR and NEP were thus 12h, 37h, 87h, 112h and 174h though some were less pronounced depending on the ocean.

**Fe -** A negative almost significant trend (p>0.05) was observed for 6 months of data (30 Sept 2011 –29 March 2012) from the MAR featuring two Fe measurements a day (at 00.00 and 12.00 UTC) (Fig. 9). Minimum and maximum concentrations were 0.25 and 2.61 µmol/l respectively with a mean of 0.98±0.43 µmol/l, which was lower than the averaged concentrations of the other deployment years (with 2.12±2.66 µmol/l averaged over 2006, 2010-2011, 2012-2013 and 2013-2014). Periodogram analyses revealed a peak at 108h (4.5 days) and a more

pronounced one at 180h (7.5 days), but none of these were significant. For the NEP, a time series of one Fe
measurement a day (at 6.00 UTC), consisting out of 4 sampling cycles, spanning >4 months was analysed (20
October 2011 – 26 March 2012). The last 49 days (31 January – 26 March 2012) were omitted due to artefacts
visible in Fig. 9, which was due to the reagents running low. Periodogram analysis of these ~3 months of data
revealed no significant periods either. Fe concentrations ranged from a minimum of 0.67 µmol/l to a maximum of
5.45 µmol/l; with mean values at 2.40±1.03 µmol/l. Mean values approached the maximum values measured by
the MAR observatory, similar to what was observed in the short-term analyses.
**Oxygen -** Due to the unresolved issues with the optodes and the oxygen concentrations measured (see section 2.2),
only the differences in overall concentration were used to describe the differences between the two sites. For the
MAR, measurements ranged from 170.54 to 251.66 µmol/l with a mean of 230.62±16.98 µmol/l. The NEP featured
distinctly lower concentrations, ranging from 23.67 µmol/l to 77.26 µmol/l with a mean of 63.42±7.15 µmol/l.
Here as well, there seemed to be more variability at the NEP than at the MAR.
**Turbidity -** Turbidity was only measured at the MAR observatory and showed several large peaks further along
in the long-term time series (e.g. during end February 2012 and May to June 2012) (Fig. 9), however none of these
observations translated themselves in the other environmental variables. There was a significant positive trend for
NTU over 9 months (p<0.001) but no significant periods were revealed by the periodogram analyses.
**4. Discussion**
**4.1. Comparison in faunal composition**
The two observatories each filmed one single assemblage over time in a limited FOV, whereas hydrothermal
edifices are characteristically inhabited by mosaics of different faunal assemblages, spatially distributed according
to local environmental conditions and microhabitats (e.g. Sarrazin et al., 1997; Cuvelier et al., 2009; 2011a,
Sarrazin et al., 2015), patterns that are enhanced by high local variability in environmental variables at centimetre
scales and steep physico-chemical gradients (Sarrazin et al., 1999; Le Bris et al., 2006). The two different study
sites also feature different spreading rates, which may influence community dynamics at vents by creating less
habitat stability in higher spreading rate settings (Tunnicliffe and Juniper 1991; Shank et al., 1998). While relative
stability in faunal composition has been observed on a number of edifices, even reaching decadal-scale stability at
some (e.g. Eiffel Tower), smaller scale variations, both in space and time, do occur (Cuvelier et al., 2011b). Hence,
the variations in faunal densities observed during this study may not apply to the hydrothermal edifice as a whole;
the presence of rhythms in the organisms and in temperature, even though observed on a smaller surface, are likely
to apply for the entire hydrothermal structure.
Vent fauna hosted by the two study sites are quite different. While there are similarities at higher taxonomic levels,
e.g. classes and families, there is only one correspondence on genus level (*Sericosura* sp., Pycnogonida) and none
on species level between both sites. A higher number of visible taxa were identified on MAR images when
compared to NEP (8 vs. 6, respectively, not taking into account microbial cover or visiting fish species). This
observation does not imply that the MAR is more diverse than the NEP since imagery only gives a partial overview
of the actual diversity (Cuvelier et al., 2012). When comparing samples, an overall higher diversity was observed
in the Pacific than in the Atlantic hydrothermal vent ecosystems, with species richness being positively correlated
with spreading rate, associated distance between vent fields and longevity of vents (Juniper and Tunnicliffe, 1997;
Van Dover and Doerries, 2005). Nevertheless, such observations remain subject to how well a certain locality is
studied and if all faunal size fractions (meiofauna to megafauna) are included in assessing diversity (e.g. Sarrazin
et al., 2015). Diversity estimates represent one of the main limitations of imagery analysis which is limited to
quantifying and correctly identifying (assessing) mega-and macrofauna (~mm). In the subsequent sections
temporal variations and behaviour (rhythms) of the separate taxa and their implications for possible microhabitat
and niche occupation will be discussed.

### 4.1.1. Engineering species

**MAR** – *Bathymodiolus azoricus* mussels visually dominate the shallow water (<2300m) vents along the MAR and
appear to be a climax community, being present for a few decades on the same edifices within the Lucky Strike
vent field (Cuvelier et al., 2011b). They form dense faunal assemblages in relatively low temperature microhabitats
(De Busserolles et al., 2009; Cuvelier et al., 2011a). A spatial segregation in mussel sizes is observed with a
decrease in size with increasing distance from hydrothermal input and corresponding thermal gradient showing
diet changes with mussel size categories (Husson et al., 2016). Contrastingly to what has been described by
Sarrazin et al. (2014), no significant interactions between mussels and other organisms were observed based on
the 6h frequency analysed here.

**NEP** – Tubeworms of the species *Ridgeia piscesae* are the main visible constituents of the filmed assemblage and
a secondary surface for the associated fauna assessed here. Their appearance rate showed a strong relationship
with the temperature recorded by probes T602 and T603 (Cuvelier et al., 2014 and this study), contrastingly to the
other taxa. Emergence/retraction movements of siboglinid tubeworms were proposed to be a thermoregulatory
behaviour or suggested to be governed by oxygen or sulphide requirements (Tunnicliffe et al., 1990, Chevaldonné
et al., 1991) or tolerance to toxic compounds (sulphides, metals, etc.). Changing hydrothermal inputs (high
sulphide concentrations/high temperature) and oxygen concentrations could thus regulate tubeworm appearances,
reflecting the tidal patterns of these environmental variables. Whilst interactions between tubeworms and other
taxa were not significantly quantifiable on the current 6h frequency of image analyses, they have been observed
and described for the hourly frequency (Cuvelier et al., 2014).

### 4.1.2. Shared taxonomic groups

**Polynoidae** – Many of the free-living polynoid species are known as active predators (Desbruyères et al., 2006)
moving rather swiftly across the FOV looking for prey and were even observed attacking extended tubeworm
plumes at NEP (Cuvelier et al., 2014). Free-living MAR scale worms were preponderantly associated with bare
substratum, while those quantified for NEP were only those observed on top or within the tubeworm bush. They
were also visible on the bare substratum surrounding the tubeworm bush but this area was not taken into account
during this study. While there was a difference in substratum association between polynoids as observed by the
two observatories, all individuals seemed to be rather territorial (see Cuvelier at al., 2014). On the MAR, one
individual appeared to repeatedly return to one single area within the FOV after excursions. Such behaviour might
be indicative of topographic memory and homing behaviour. The Atlantic commensal polynoid *Branchiplynoe*
*seepensis* can occasionally be observed outside of the mussel shells (Sarrazin et al., 2014), wherein it normally
resides, but not on the image sequence analysed here.

**Gastropoda** – Buccinid (NEP) and bucciniform (MAR) gastropods appeared more related to less active
environments. Both species are considered predators or scavengers (Desbruyères et al., 2006; Martell et al., 2015).
Within the MAR setting, snails (*Phymorhynchus* sp.) were present in very low abundances (1 or 2 individuals at
most) and were positioned on bare rock with no fluid flow. In the NEP setting, whelks (*Buccinum thermophilum*)
were generally more abundant on areas inhabited by vent animals. No correlation with emerging fluid temperatures
was observed nor was a substratum preference revealed (Martell et al., 2015). Abundances observed within the
FOV tended to vary from 1 to 6 individuals, while they were shown to congregate in groups of 5 or more
individuals at MEF (Martell et al., 2015).

**Pycnogonida** – Sea spiders showed a very distinct spatial distribution at NEP featuring a localised clustering
behaviour (see heat maps published in Cuvelier et al., 2014), whilst their presence on the MAR was occasional.
MAR pycnogonid individuals were only observed visiting the edge of the mussel bed which was further away
from the fluid exit. A large difference in pycnogonid densities (ind/m$^2$) was observed between the two sites as
well, with a ratio of 1/250 MAR vs. NEP. Increased activity and aggregations of more than 5 individuals (and
increased intra-species contact) at NEP were linked to conditions of high temperature-low oxygen saturation
(Lelièvre et al., 2017). Interestingly, these organisms all belong to the same genus, namely *Sericosura*. The species
known for the Lucky strike vent field (MAR) is *Sericosura heteroscela* while there are multiple species (within
the same genus) for the Main Endeavour Field (NEP) among which *Sericosura verenae*. All *Sericosura* species
from the Ammotheidae family known so far appear to be mostly obligate inhabitants of hydrothermal vents or
other chemosynthetic environments (Bamber, 2009). While being an abundant taxon with a localised clustering
behaviour at the NEP site, it is scarce and vagrant at the MAR. Their microhabitat and niche occupation at the
studied sites is likely to differ, causing the discrepancies observed.

**Microbial cover** – This is a generic term used to refer to the microbial mats colonising various surfaces in the
vent environment without assuming similar microbial composition. While no significant relationships were
revealed between microbial cover and fauna for NEP in the current study, a significant negative correlation was
observed for this site between pycnogonids and microbial cover based on the same imagery analysed with a higher
frequency (4h instead of 12h), which was attributed to pycnogonid grazing (Cuvelier et al., 2014). For MAR,
significant negative correlations existed between microbial coverage and mussels and microbial coverage and
shrimp. For the mussels, this could be due to scattering and repositioning of individual mussels: as mussel
reposition on top of the microbial mats, they decrease the visible and assessable microbial coverage. The negative
relationship between shrimp and microbial cover could be caused by the shrimp grazing on microorganisms
(Gebruk et al., 2000; Colaço et al., 2002; Matabos et al., 2015).

### 4.1.3. Regional taxa

**MAR**

**Alvinocaridid shrimp** – The hydrothermal shrimp observed by the MAR observatory mostly belong to the *Mirocaris fortunata* species. On the images analysed, they were most abundant in the main axe of flux. Matabos et al. (2015) quantified this to about 60% of the shrimp abundances (to 69cm of an emission), confirming previous distributional patterns of shrimp being indicative of fluid exits and characteristic for warmer microhabitats (Cuvelier et al., 2009, 2011a; Sarrazin et al., 2015). Their thermal resistance and tolerance corroborates this pattern (Shillito et al., 2006). Because their distribution is linked to the presence of fluid exits and flow, a significant positive correlation between shrimp and temperature would be expected. To date however, such a relationship could not be designated, not in this study or in previous studies based on data from the deep-sea observatories (Sarrazin et al., 2014; Matabos et al., 2015), though Sarrazin et al. (2014) did show a significant positive correlation between *Mirocaris fortunata* abundances and vent fluid flux.

**Bythograeidae (Decapoda)** – *Segonzacia mesatlantica* crabs were mostly associated with the mussel beds and anhydrites, as where the shrimp (Matabos et al., 2015). Some interactions between crabs and shrimp were observed mostly resulting in shrimp fleeing. Possible significance of these interactions (mostly territorial in nature) were described in more detail by Matabos et al. (2015).

**Bythitidae (Osteichthyes)** - The fish *Cataetyx laticeps* was frequently observed at the base of the Eiffel Tower edifice within the Lucky Strike vent field (Cuvelier et al., 2009). No feeding on the benthic hydrothermal fauna was observed during the 6h frequency image analyses.

**NEP**

**Majidae (Decapoda)** - Contrastingly to the 1h frequency observations (Cuvelier et al., 2014), no spider crabs were observed visiting the filmed assemblage on a 6h frequency imagery analyses. Whilst this majid spider crab is known as a major predator at hydrothermal vents, no such actions were recorded by our observatory module.

**Zoarcidae (Osteichthyes)** – Similarly to *Cataetyx* fish on the MAR, no visible activities of feeding or predation of *Pachycara* sp. eelpouts were observed on the NEP. Cuvelier et al. (2014) proposed that the eelpouts (and fish in general) may be more sensitive to the effects of lights but this hypothesis, based on behavioural observations, could not be confirmed in the present study due to the low-resolution observation frequency.

### 4.2. Short term variations and rhythms in fauna and environment

When looking at the engineering taxa for each ocean, a clear diurnal rhythm was observed in visible (i.e. out of their tubes) tubeworms (NEP), while there was a lack of temporal rhythms in mussel densities (MAR). However, taking in to account the characteristics of both chemosynthetic taxa, counts of mussels with open valves and extended siphons instead of densities should be used for comparison to tubeworms outside their tube. This difference in assessment could account for the lack of temporal periodicities at the MAR, where mussel valve openings or visible siphons were impossible to quantify due to the larger distance between the observatory and the

filmed assemblage. Different causes might trigger a mussel to open his valve or a tubeworm to come out of its
tube and these can be either attributed to an external trigger (e.g. retraction or closure after possible predation
actions (for tubeworms: Cuvelier et al., 2014; for mussels: Sarrazin et al., 2014)) or to their physiology (need for
nutrients or saturation). No significant links have yet been established between fluid flow and open mussel valves
(Sarrazin et al., 2014) but some indications of tidal rhythmicity were visible (Matabos et al., unpublished data).
No consistent statistically significant link between fluid flow and tubeworm appearance has been revealed to date
either (Cuvelier et al., 2014), although a steady significant semi-diurnal tidal rhythm over time was observed. The
niche occupation and role within the ecological succession over time of mussels and tubeworms are very different
for the two oceans. In Pacific monitoring studies, tubeworms are out-competed by mytilid mussels when
hydrothermal flux start to wane (Hessler et al., 1985; Shank et al., 1998; Lutz et al., 2008; Nees et al., 2008), while
the latter appear to represent a climax community in the more stable Atlantic <2300m (Cuvelier et al., 2011b).
Nevertheless, 23 days appears too short to allow observation of succession patterns.

Next to the engineering species, only a few other taxa showed significant periodicities in densities over time,
namely polynoids for MAR and NEP, and buccinids for NEP. The lack of significant periodicities in MAR shrimp
was corroborated by a long-term study by Matabos et al. (2015). Both polynoids and buccinids displayed multiple
day periodicities instead of tidal cycles, which could be mostly reduced to harmonics of tidal cycles that become
more visible further along in the time series as they become more pronounced over time. For both taxa, the multiple
day periodicities approached those visible in Fe, i.e. 4.5 and 7.5 days (though non-significant) and besides an
apparent preference for lower temperatures, there were no significant links with temperature (as corroborated by
Lelièvre et al. (2017) for the polynoids). Additional high resolution investigations will be necessary to corroborate
or validate these observations. Overall, the reasons for the lack of periodicities in fauna can be twofold: either the
taxon in question is unevenly represented in low abundances and therefore too heterogeneous (rendering any
statistical test difficult which was the case for MAR crabs and pycnogonids) or the recording/analysing frequency
does not allow discerning of significant periods. The shortest period to be resolved is twice the interval between
the observations of a time series. Hence, caution is needed when interpreting patterns as the recording and
analysing frequency influences observations. A previous higher resolution study (hourly frequencies) already
showed that depending on the frequencies investigated the type of relationships (significance, positive or negative)
between the taxa might change (Cuvelier et al., 2014).

While certain environmental variables might explain a large amount of variation occurring in a single taxon (e.g.
NEP tubeworm appearances and temperature from probes T602 and T603), a wider variety of environmental
variables measured at multiple sampling points across the FOV in a resolution similar or higher than the imagery
analyses frequency should be considered in order to explain and comprehend the whole of community dynamics.
This was also illustrated with the temporal split groups identified in community composition constrained by time,
where the predictive power of the split groups was rather low and groupings could not be corroborated with
changes in the environmental variables. Split groups were quite similar for the larger groups (those with higher n)
with those at the MAR occurring 6 hours later than those at the NEP. A slower pace in significant detectable
changes in overall faunal composition in the Atlantic vs. the NE Pacific could be explanatory. For instance,
difference in spreading rate was shown to be directly proportional to different rates of change in community
dynamics between slow-spreading MAR and faster-spreading NEP (Cuvelier et al., 2011b).

**4.3. Long term environmental variations and rhythms**
At hydrothermal vents, temperature is a proxy of sulphide and Fe concentrations and most importantly of the
hydrothermal vent input. Highest minimum temperatures were recorded at the MAR where the probe was
positioned closer to a visible fluid exit, whereas NEP temperatures were more variable and displayed broadest
ranges. It is important to bear in mind that ambient seawater temperature at 1700m on the MAR is higher than that
at 2200m depth in the NEP (4°C vs. 2°C respectively). When taking this into account and rescaling the temperature
values, mean and maximum temperatures were highest at NEP. Highest positive and significant autocorrelation
values indicated a ~5-6h lag between MAR and NEP, with MAR leading. Interestingly, the hour difference
between the two sites corresponds to ~6 hours as well. The geographical distance separating the two localities does
thus not only allow to quantify the time difference between two sites but also the delay in the tidal rhythms
observed between the two.

Tidal rhythms were discernible in both NEP and MAR temperature series. Potential mechanisms causing tide-
related variability in hydrothermal fluids included the modulation of seafloor and hydrostatic pressure fields by
ocean tides, modulation of horizontal bottom currents by tides and solid earth tide deformations (Schultz and
Elderfield, 1997; Davis and Becker, 1999). For NEP, diurnal periods at ~25 h were discerned for both temperature
probes (T602 and T603). Significant semi-diurnal periods were also found in T602, though for T603 they could
only be identified based on their harmonics. The MAR temperature time series also had a distinguishable semi-
diurnal component. Tidal rhythms observed in the temperature time series for NEP and MAR were concordant
with observed tidal signals for the respective regions. For instance, in the North-East Pacific, measured tides in
the Barkley Canyon, another instrumented node from ONC closer to shore, were mixed semidiurnal/diurnal at
870m depth (Juniper et al., 2013). In the same canyon, periods of enhanced bottom currents associated with diurnal
shelf waves, internal semidiurnal tides, and also wind-generated near-inertial motions were shown to modulate
methane seepage (Thomsen et al., 2012). While, temperature variability at hydrothermal vents at Cleft Segment
on the Juan de Fuca Ridge was shown to greatly diminish when current directions did not shift in direction with
the tides, it was suggested that the modulation of temperature by tides was only indirect, through the modulation
of horizontal bottom currents (Tivey et al., 2002). These horizontal bottom currents showed 12.4h tidal periodicity
which was also found in the temperature time series of the aforementioned article as well as in our NEP temperature
time series. Consistent with the main orientation of the ridge and the topography of Grotto, temperature and oxygen
saturation at the NEP deployment site were shown to be strongly and significantly influenced by the northern and
southern horizontal bottom tidal currents (along the valley axis) (Lelièvre et al. 2017). Patterns in temperature
variation of the MAR time series corresponded to the tidal signal observed in the Lucky Strike vent field at 25h
and to the semi-diurnal tidal oscillation at 12h30 (Khripounoff et al., 2000; 2008).

Between oceans, differences were observed in tidal rhythms of high (>200°C) and low (<10°C) temperature
records. For the NEP, the tidal influence appeared to wane in high temperature records making tidal signals less
clear or even non-existent (Tivey et al., 2002; Hautala et al., 2012). While for the MAR the semi-diurnal variability
in the high temperature records was shown to be more significant and to be more coherent with pressure than those
observed in low-temperature (Barreyre et al., 2014). Unfortunately, we cannot corroborate this with the current
study as only low-temperature time series were recorded by both ecological observatories. Even though we
revealed some similarities in the rhythms of MAR and NEP low temperature series collected for the same period,
there were indications that local hydrography and associated bottom-currents play a major role on the temporal
variability of diffuse outflow and vent discharges (Barreyre et al., 2014, Lee et al., 2015). Clear peaks in
temperature variables were noticeable at ~6-7 days in MAR and NEP. We do not know what caused this period to
be significant. In comparison, at Cleft Segment more southwards on the Juan de Fuca Ridge (NEP), Tivey et al.
(2002) found 4-5 day broadband peaks in temperature from diffuse flow as well as high-temperature vents which
were thought to be storm-induced from the sea-surface.
Fe is commonly used as a proxy for vent fluid composition. Higher Fe concentrations would thus be expected
where temperatures were higher, in this case at MAR (vs. NEP). However, the opposite was observed here. The
Fe concentrations reported here for the MAR were lower than the Fe concentrations from other deployment years
at the same site (Laes-Huon et al., unpublished data). The 2011 concentrations recorded at the MAR were close to
the detection limit of the CHEMINI instrument (0.3 μmol/l). Additionally, the MAR system was not calibrated *in
situ*, contrastingly to the NEP, which could have generated a lower accuracy in the calculated concentrations,
though question remains if such large discrepancies can be explained by this feature alone. The location of the
sample inlet and the high spatial variation occurring at hydrothermal vents might contribute to the patterns
observed. The values observed at NEP were in the same order of magnitude as those reported for the Flow site
also on the Juan de Fuca Ridge (i.e. 0 to 25 μmol/l, Tunnicliffe et al., 1997). No significant periods (based on 12h
or 24h recording frequency) were found at the sites for the duration of the deployment, although some indications
of 4.5 and 7.5 day periodicities could be observed at the MAR and 3.8 day cycles for Fe concentrations were
detected in the same sampling area for 2012-2013 (Laes-Huon et al., in press). For the North East Pacific, 4 day
oscillations in currents near seamounts along the crest of the Juan de Fuca Ridge were observed (Cannon and
Thomson, 1996), however, these were not visible in the Fe time series at NEP, although 4.5 day periodicities were
visible in buccinids and polynoids (Cuvelier et al., 2014). Hence, there were some indications of multiple day
periodicities, but these findings need to be corroborated, preferably by using a higher sampling frequency.
Turbidity (NTU) levels observed showed several large peaks over time. Particle flux at Lucky Strike combines
both large and small diameter particles which have different settling velocity (Khripounoff et al., 2000).
Kripounoff et al. (2008) showed an increased particle flux in April that reached a maximum end May (2002).
These do not correspond to the peaks observed here (in this study peaks were most pronounced at the end of
October, February to March and May to July) but turbidity peak occurrences tend to differ between years and
seasons. Due to seasonal peaks, longer time series will be needed to reveal recurrent patterns.
Generally, multiple day periodicities were harder to reveal as many of them can be reduced to harmonics of the
tidal cycles. In this perspective, the long(er)-term environmental variable analyses were considered more robust
due to increased number of data points. Nevertheless, there is not much we can currently say on multiple day or
hebdomadal cycles observed in the time series presented here.

**4.4. Limitations**
Overall at hydrothermal vents, it remains hard to establish relationships among the environmental variables
measured *in situ*. Ratios of temperature to chemical concentrations are not constant, and can vary between sites
(Le Bris et al., 2006; Luther et al., 2012). There is also the issue of high variance (and noise) in environmental
variable time series as well as that of a possible delay in appearance of certain peaks, which makes it difficult to
unravel patterns. Such a delay between environmental variable recordings might exclude the ability of
unravelling/exposing correlations. The example for Fe and temperature recordings, where a delay of 1 to 5 min
precluded a direct correlation for each sample point, was presented by Laes-Huon et al. (2016).

Caution is needed when programming the recording frequencies of imagery and environmental variables. Despite
being mainly restricted by battery life (wireless observatories), light usage (wired observatories) or quantity of
reagents (both), a 6h analysing frequency might not be the most representative to assess faunal variations and links
with the environment. Indicative of this are the differences observed when analysing different frequencies as
briefly touched upon in Cuvelier et al. (2014) and comparing them with those presented here. It still proves difficult
at hydrothermal vents to link faunal variations with single-point environmental variables measured *in situ*. This
can be attributed to the high spatial and temporal variation of the environmental gradients compared to the larger
FOV assessed and to the recording frequencies or complexity of *in situ* measurements with corrections to be
applied and possible delays. Temperature still seems the best proxy for faunal variations, however not all faunal
presences/absences, abundances or the entirety of community dynamics can be explained solely by temperature.
Biotic interactions are at play as well. While these can be observed thanks to the remote observatory set-up, long-
term high resolution data need to be assessed (Matabos et al., 2015).

The influence of the lights on the fauna was hard to discern during this study, though supposedly fish presence
would be more impacted when compared to invertebrate fauna (Aguzzi et al., 2010; Cuvelier et al., 2014).

Deployment of probes has also proven to be a predicament. While more accessible sites tend to be preferred and
selected, deployment setting, accessibility, underwater conditions (e.g. currents), ROV manoeuvrability and
piloting skills also influence the final observatory set-up.
**5. Conclusions**
Influence of the tides is visible in both settings, most clearly in temperature variables and in tubeworms
appearances. The geographical distance separating the two localities is shown to not only quantify the time
difference between two sites but also the delay in the tidal rhythms observed in temperature values (which is at a
~6h lag) between the MAR and NEP. Temporal split groups in community composition are rather similar between
both settings, though the 6h delay is visible as well. Shared taxa comprised one genus (*Sericosura*), one family
(Polynoidae) and one class (a buccinid and a bucciniform Gastropoda) and based on their relative abundance and
behaviour, they seem to occupy different niches at the different hydrothermal vents. Nevertheless, it remains
complicated to unravel links with the environment and to discern which environmental variable is the most
influential or explanatory. To date, temperature remains the most explanatory, though it cannot explain the entirety
of community dynamics. This is likely due to the high spatial variation at hydrothermal vents and the single point
measurements done by the environmental probes. A persistent need remains for more complementary and
representative data, measured at frequencies similar or higher than the imagery recordings and at multiple points
in the FOV. Recording frequencies are crucial: a 6h recording frequency might not be good enough to represent
the *in situ* reality. Also the implementations of instruments that do not imply complex tools but allow the
assessment of additional environmental variables (e.g. current meters) could be a way forward. (Semi-) Automated
tools should be developed for specific taxa and settings to assist in assessing faunal abundances in images.

**Acknowledgments**
We thank the captains and crews of the R/V Pourquoi pas? and the R/V Thomas G. Thompson for their steadfast
collaboration in the success of the MoMARSAT and Neptune Ocean Networks Canada cruises. We are grateful to
the Victor6000 and ROPOS ROV pilots for their patience and constant support. The authors would like to thank
the LEP technical team for its valuable help both at sea and in the lab, the TEMPO and TEMPO-mini engineers
and technicians who developed and maintained the modules and the Ocean Networks Canada scientists and
engineers, whose engagement and professionalism made this study possible. DC is supported by a post-doctoral
scholarship (SFRH/BPD/110278/2015) from FCT and this study had also the support of Fundação para a Ciência
e Tecnologia (FCT), through the strategic project UID/MAR/04292/2013 granted to MARE. Thanks to IMAR for
support to DC during the preparation of this manuscript. This project is part of the EMSO-Açores research program
that benefited from an ANR research grant (ANR LuckyScales, ANR-14-CE02-0008).

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

**Figures**


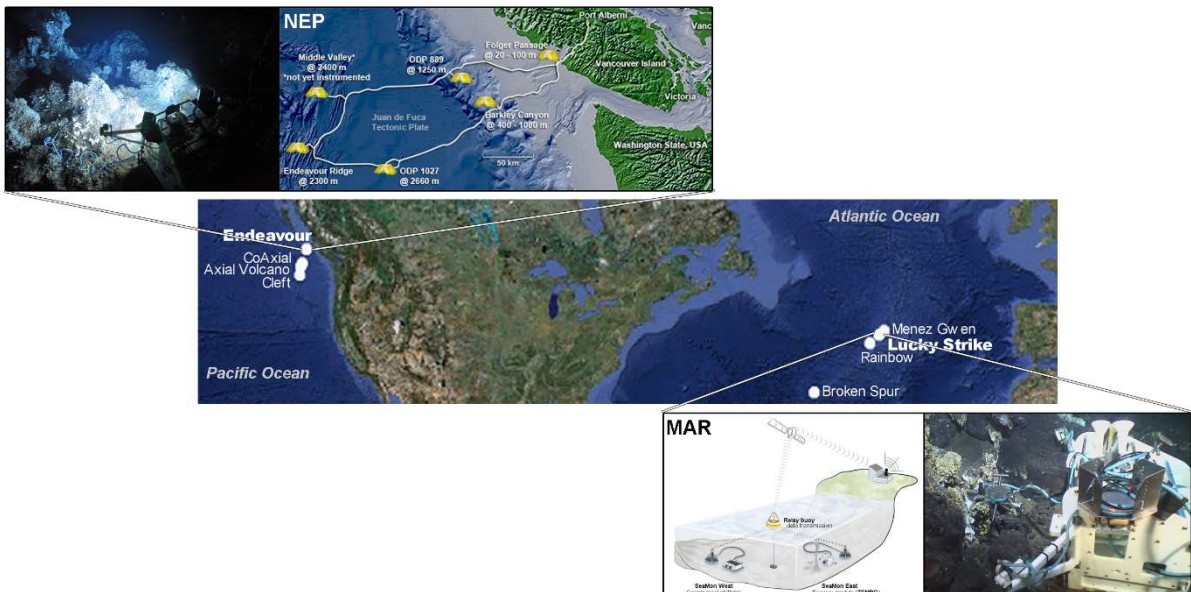


**Fig. 1 Location of the two study-sites in the Atlantic and the Pacific Ocean, along with some other well-known vent fields for reference purposes. The NEP inset (top) shows the location of the different instrumented nodes of Ocean Networks Canada at the right and the TEMPO-mini ecological module deployed at Main Endeavour Field on the Juan de Fuca Ridge. The MAR inset (bottom) represents a sketch of the Atlantic observatory (EMSO-Açores) at Lucky Strike vent field on the left and the TEMPO ecological module on the right. For more details of the exact location of the observatories within the hydrothermal vent fields see Matabos et al. (2015) for MAR and Cuvelier et al. (2014) for NEP.**

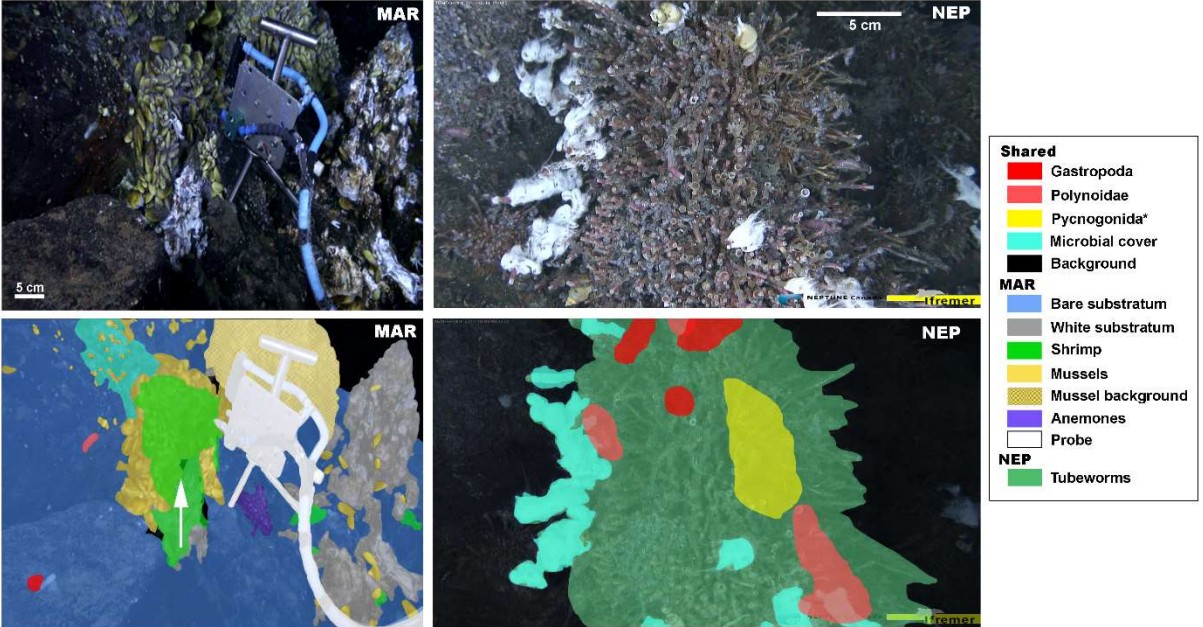

**Fig. 2. Sample image recorded by the ecological observatory modules for MAR and NEP (top) and a map of the fields of view (FOV) featuring the various taxa assessed (bottom). Taxa or other features that are shared between the two observatories share the same colour codes. Gastropoda applies to Buccinidae for NEP and bucciniform Turridae on MAR. White substratum is possibly anhydrite with encrusted microbial mats. 'Mussel background', 'background' and 'probe' were areas that were not assessed. The white arrow represents the fluid flow exit and direction. No visible emission was observed on NEP. Visiting fish and crab species were not included (Table 2). Crab presence on MAR tends to correspond predominantly to shrimp distribution (Matabos et al., 2015). Surfaces filmed and analysed are listed in Table 2. '*' is a shared taxon but not visible on MAR sample image or map due to the scarce presence and low densities.**

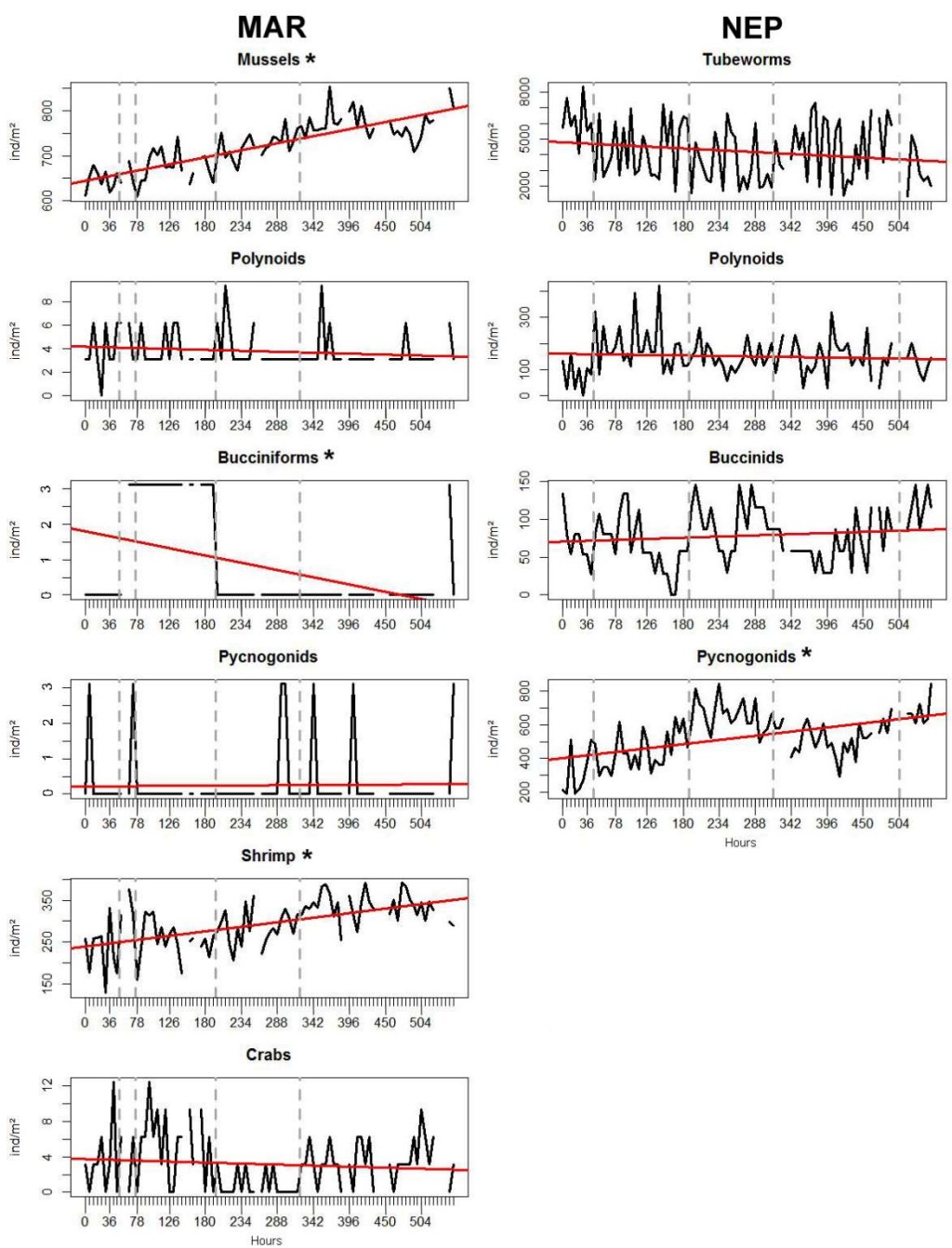

**Fig. 3. Temporal variations in faunal densities for MAR and NEP along with trend lines (in red) and MRT temporal**
**groups (grey vertical dotted lines), x-axis show the sampling frequency every 6h. Taxa with significant trends (p<0.05)**
**are marked with an \*.**

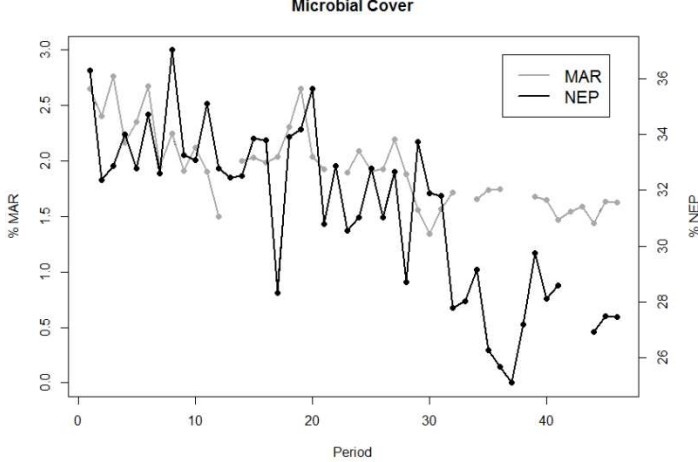

**Fig. 4. % Microbial cover every 12h, for the imagery period analysed. X-axis contains periods, 1 period=12h**

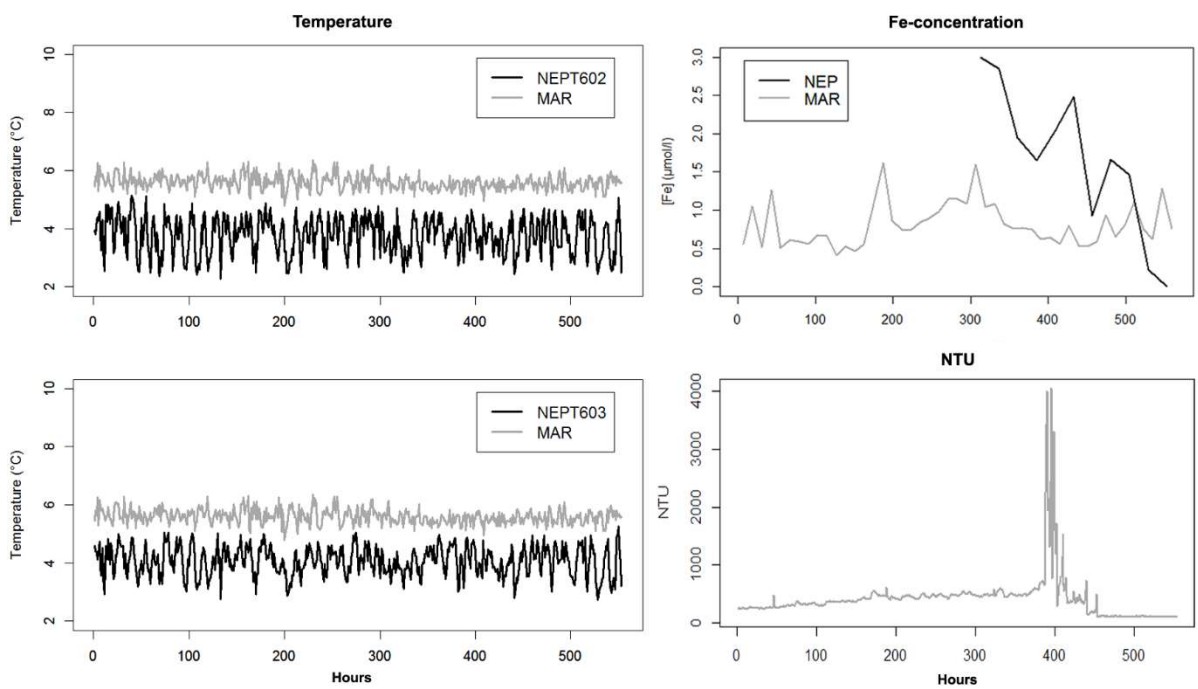

**Fig. 5. Short term environmental variables (23 days) averaged per hour during the imagery analysis period. Variables**
**measured at both deployment sites are presented in the same graphic (temperature and Fe). Fe has a daily frequency**
**for the MAR but a 12h frequency for the NEP and recording times differ. NTU (Turbidity) was only available for the**
**MAR.**

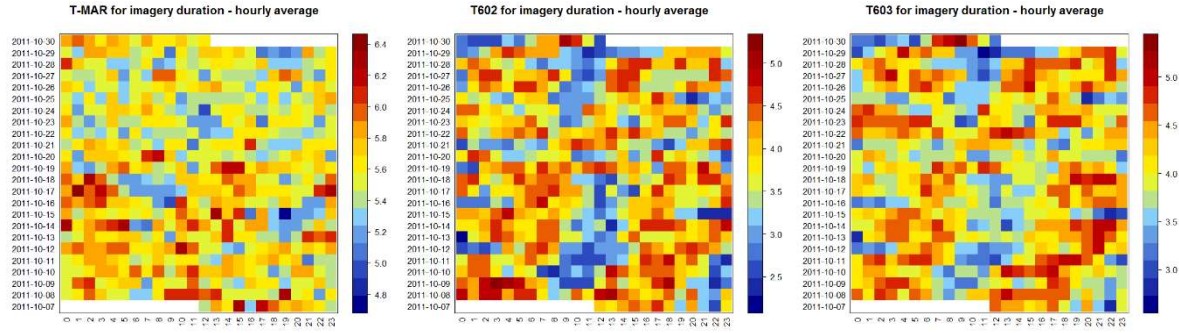

**Fig. 6. Hourly temperature values (°C) for T602 and T603 probes from NEP and the MAR temperature probe. Red**
**are higher temperatures while blue are lowest temperatures. Dates correspond to the duration of the imagery analyses**
**(23 days).**

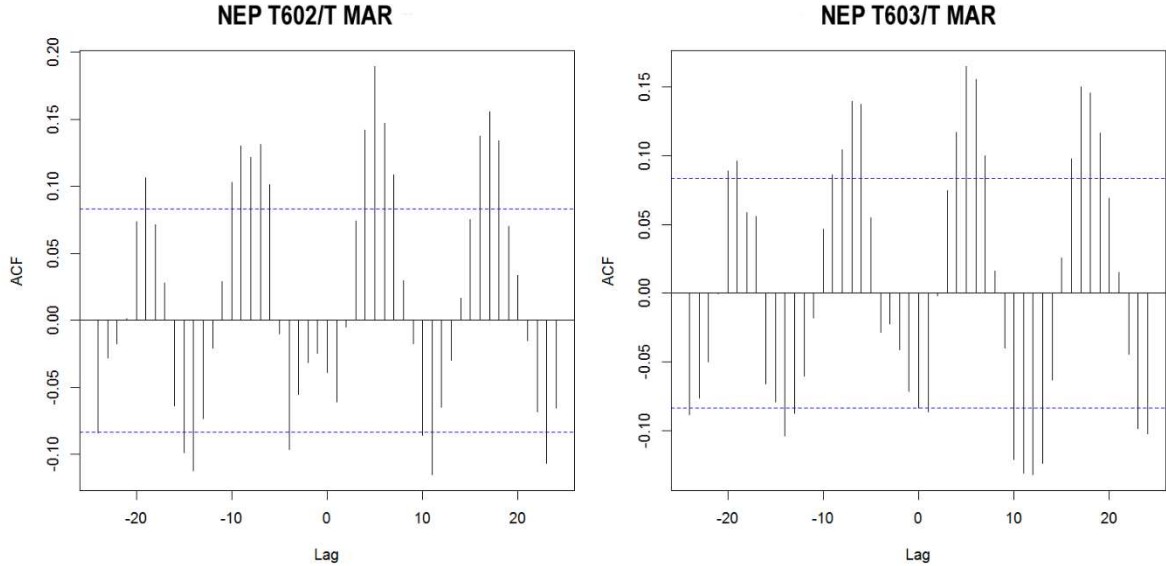

**Fig. 7. Cross correlations of the hourly temperature values. ACF=autocorrelation function on y-axis, 1 lag equals 1**
**hour on x-axis. Comparisons are made between the MAR probe (T MAR) and T602 (NEP T602) on left side and**
**MAR (T MAR) and T603 (NEP T603) on the right. The horizontal dashed lines indicate the point of statistical**
**significance at ACF=0.8, with the lines above towards 1 and below towards minus 1 being significant.**

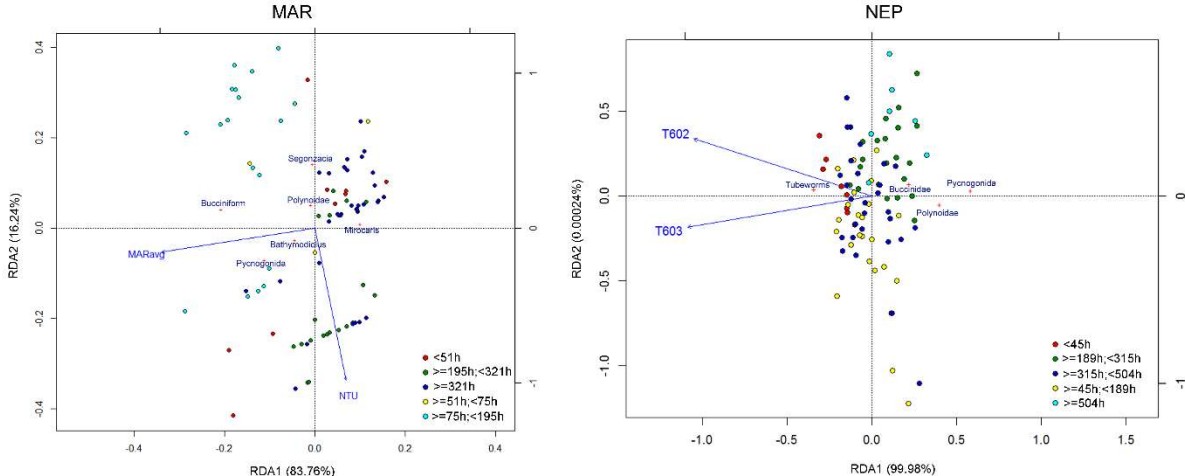

Fig. 8. Redundancy Analysis (RDA) ordinations featuring Hellinger transformed faunal densities and environmental variables both at a 6h frequency. MARavg is the temperature time-series from the MAR and NTU is turbidity. T602 and T603 were the NEP temperature probes. Temporal splits groups were colour-coded in the ordination plots.

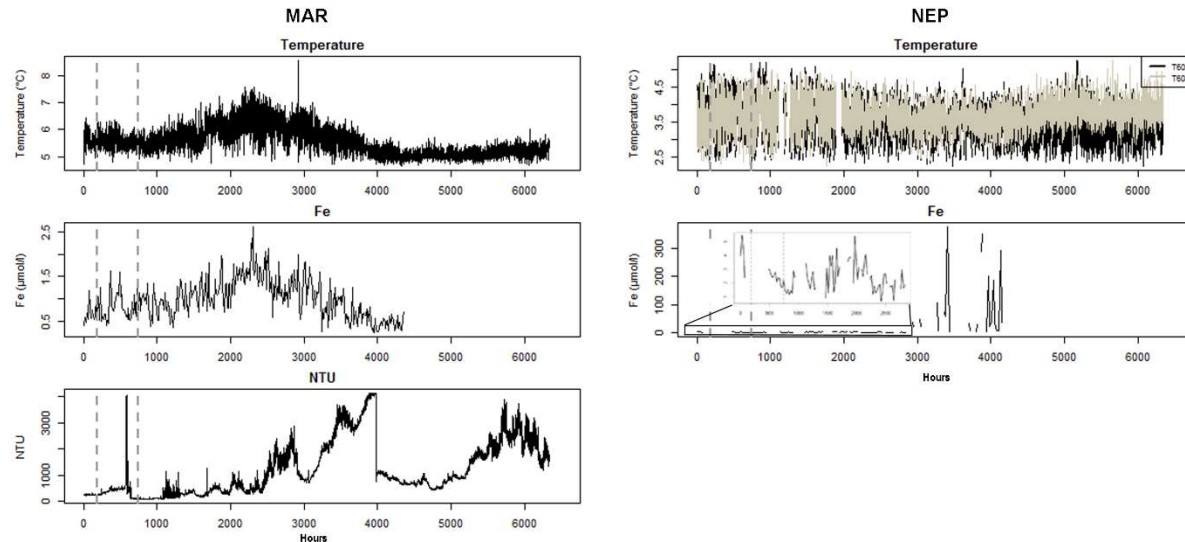

Fig. 9. Long-term environmental variable overview. Temperature time-series at MAR and NEP represent hourly temperature data spanning 9 months. Fe was recorded during 6 months, twice a day at MAR and daily at NEP. Dotted vertical lines delineate the period for which the images have been analysed. Inset box in Fe graph for NEP shows variation occurring during the first 4 months in more detail.

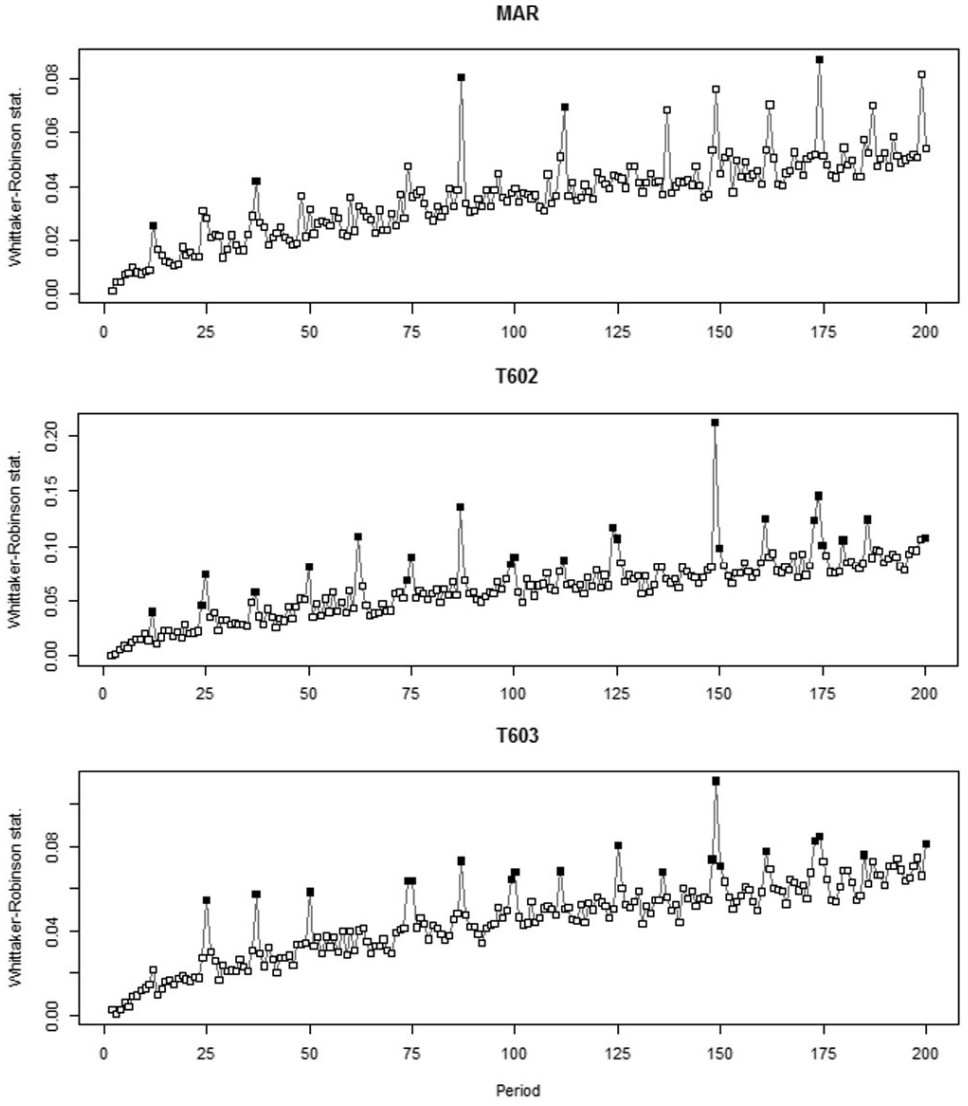

**Fig. 10. Periodogram analyses of ~9 months of hourly temperature measurements for MAR and NEP (T602 and T603)**
**represented as a one-week period (equalling 200h). 1 period=1h. Black squares indicate periods significant at the 5%**
**level.**

# Appendix/Supplementary figures

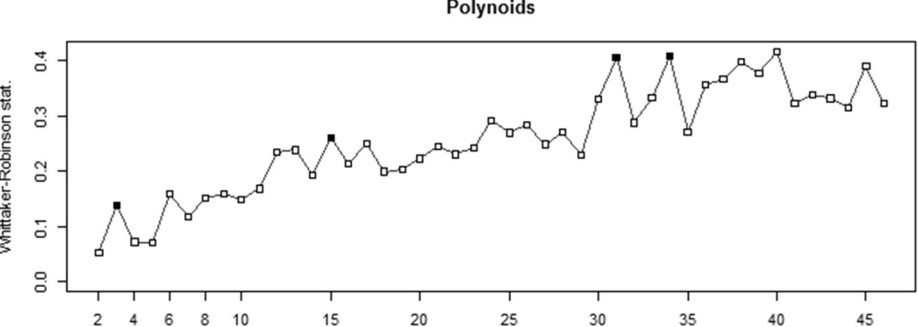

**Fig. S1. MAR faunal periodogram on polynoid densities with a 6h frequency (1 period on x-axis=6h) of 23 days, all other taxa had no significant periodicities and were thus not shown. Black squares indicate periods significant at the 5% level.**

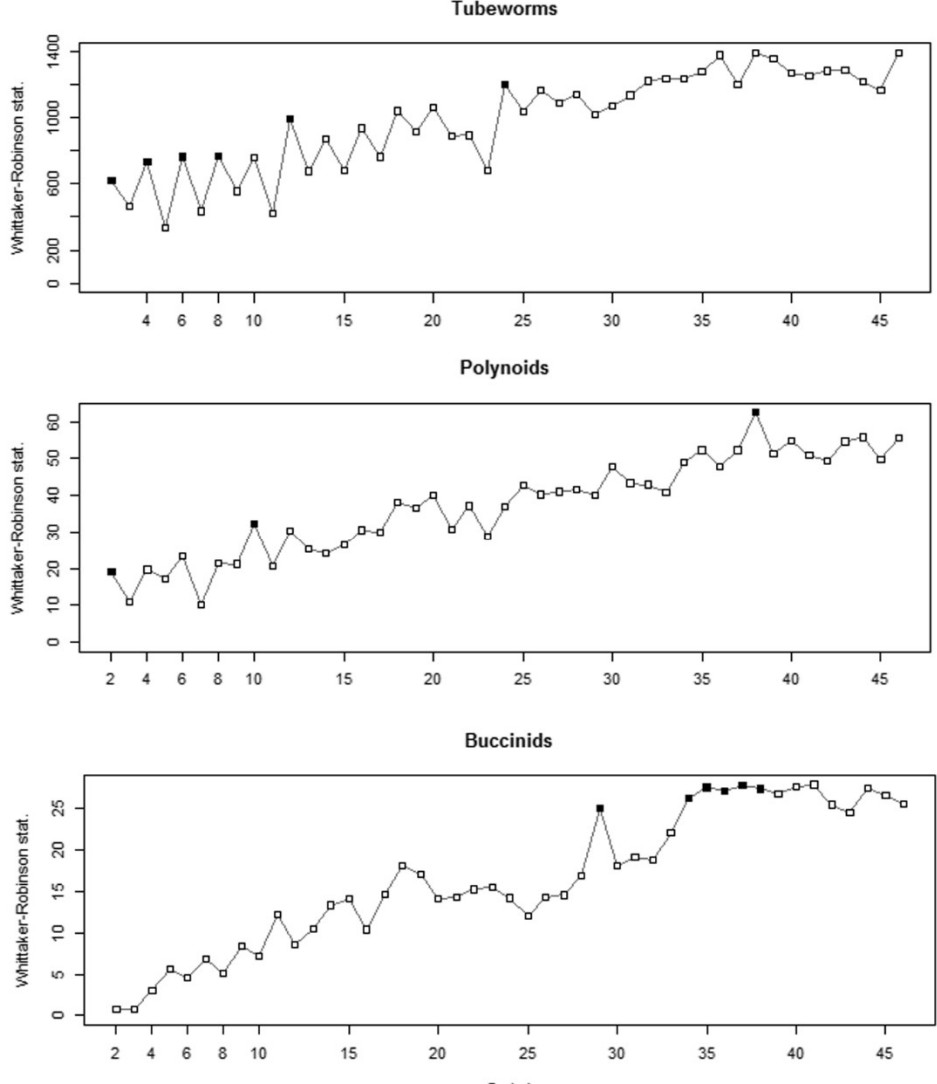

**Fig. S2. NEP Faunal periodograms of 23 days featuring significant periodicities. Taxa presented are tubeworm, polynoid and buccinid densities with a 6h frequency for the MAR (1 period on x-axis=6h), pycnogonids showed no significant periodicities and were not shown. Black squares indicate periods significant at the 5% level.**

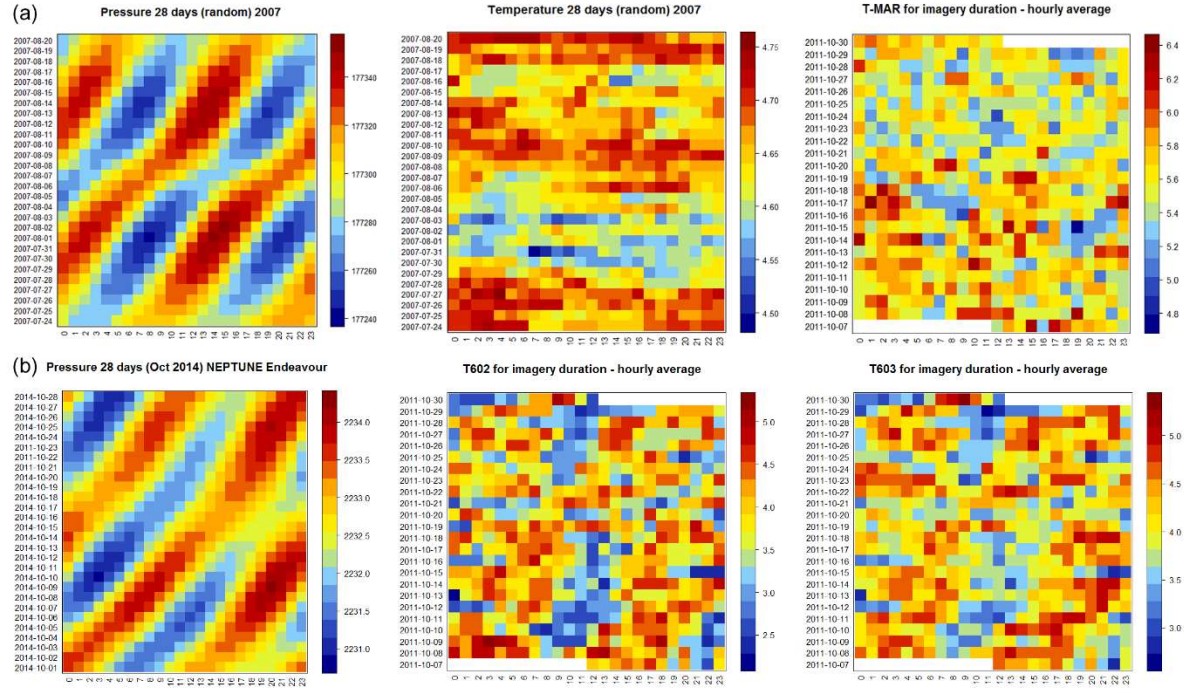

**Fig. S3. Comparison of cyclicity in pressure data and temperature for (a) MAR and (b) NEP Red are higher values while blue are lowest values. Pressure data for MAR originates from 2007-2008 and was recorded at Seamon West of the EMSO-Azores observatory and represents a random 28 day (lunar) period (data courtesy of Valerie Ballu). Pressure data for NEP were downloaded from ONC Portal from the BPR (NRCan Bottom Pressure Recorder deployed at MEF/Endeavour) ("Ocean Networks Canada Data Archive http://www.oceannetworks.ca, Total Pressure data from 1-29 Oct 2014, University of Victoria, Canada, Downloaded on 16 Jun 2015"). A random selection of 28 days in October 2014 is presented here (no earlier data were available).**

# **Tables**

**Table 1: Overview of the location, data recorded and the recording resolutions of all variables of the two observatories on the NEP and MAR.**

| | **TEMPO MoMAR/EMSO-Açores (MAR)** | **TEMPO-mini NEPTUNE (NEP)** |
|---|---|---|
| Energy provision | **Batteries (Wireless)** | **Cabled** |
| Coordinates Lat<br>Coordinates Long | N 37° 17.3321'<br>W 32° 16.5334' | N 47°56.9574'<br>W 129°05.8998' |
| Depth | 1694 m | 2168 m |
| **Imagery** | 4 min every 6h (at 0.00, 6.00, 12.00, 18.00 UTC) | Continuous for ~23 days followed by 30 min every 4h (at 2.00, 6.00, 10.00, 14.00, 18.00 , 22.00 UTC) |
| **Temperature** | 1 measurement every 5min | 1 measurement every 30sec |
| **Optode (oxygen + temperature) *** | 1 measurement every 15min | 1 measurement every 15min |
| **Chemini Fe** | Twice a day | Twice a day/daily |
| **Turbidity (NTU)** | 1 measurement every 15min | NA |

**\*limited usefulness due to issues related to correctly calculate the oxygen concentrations.**

Table 2: Overview of the characteristics of the images analysed such as surface covered and taxa assessed within the
FOV. The analysed surface on the MAR is about 10 times larger than that on the NEP. Gaps are failed or unusable
video recordings.

| | TEMPO MoMAR (MAR) | TEMPO-mini NEPTUNE (NEP) |
|---|---|---|
| # Images (6h frequency) | 84 (93 total with 9 gaps) | 88 (93 total with 5 gaps) |
| Surface filmed | ~0.3802m² (ca. 52.8 x 72cm) | ~0.0661m² |
| Surface analysed (see fig. 2) | ~0.322m² | ~0.0355m² (ca. 20 x 18cm) |
| **Taxon densities** | | |
| **Annelida** | | |
| Siboglinidae | NA | *Ridgeia piscesae* |
| Polynoidae | Multiple species (Desbruyères et al., 2006) | Multiple species (Cuvelier et al., 2014) |
| **Arthropoda** | | |
| Alvinocarididae | *Mirocaris fortunata* | NA |
| Bythograeidae | *Segonzacia mesatlantica* | NA |
| Majidae | NA | *Macroregonia macrochira* * |
| Pycnogonida | | |
| Ammotheidae | *Sericosura heteroscela* | Among others: *Sericosura verenae* |
| **Cnidaria** | | |
| Actiniaria | Anemones sp. | NA |
| **Echinodermata** | | |
| Ophiuroidea | Ophiuroid sp. | NA |
| **Mollusca** | | |
| Buccinidae | NA | *Buccinum thermophilum* |
| Limpets (Lepetodrilidae, Provannidae etc.) | NA | Multiple species |
| Mytilidae | *Bathymodiolus azoricus* | NA |
| Turridae | *Phymorynchus* sp. (bucciniform) | NA |
| **Pisces** | | |
| Bythitidae | *Cataetyx laticeps** | NA |
| Zoarcidae | NA | *Pachycara* sp.* |
| **Surface coverage** | % Microbial mats (12 h frequency) | % Microbial mats (12 h frequency) |

* Are visiting predators.

**Table 3. Temporal split groups for MAR and NEP based on MRT analysis. n=number of images**

| MAR | | NEP | | Timespan |
|---|---|---|---|---|
| <51h | n=9 | < 45h | n=8 | ~ 2 days |
| ≥ 51h, <75h | n=3 | ≥ 45h, < 189h | n=24 | > 2 days, < 8 days (spanning ca. 6 days) |
| ≥ 75h, < 195h | n=18 | | | |
| ≥ 195h, < 321h | n=20 | ≥ 189h, < 315h | n=21 | >8 days, < ~ 13 days (spanning ca. 5 days) |
| ≥ 321h – 553h | n=34 | ≥ 315h, < 504h | n=28 | > ~13 days, <21 days for NEP (spanning ~8 days) > ~13 days, 23 days (10 days for MAR) |
| | | ≥ 504h – 553h | n=7 | > 21 days till end of recordings (~ 2 days) |



**Table 4. Mean, maximum and minimum temperatures as measured by the probes and, for comparison purposes**
**rescaled to ambient seawater temperature (highlighted in grey). See Fig. 5 for significant differences in raw temperature**
**values. Variance and standard-deviations are presented as well. Bold values represent highest values which tend to**
**change if rescaled to ambient seawater temperature or not.**

| | Mean (°C) | | Max (°C) | | Min (°C) | | Var | Stdev |
|---|---|---|---|---|---|---|---|---|
| MAR | **5.59** | 1.59 | **6.36** | 2.36 | **4.79** | **0.79** | 0.066 | 0.258 |
| NEPT602 | 3.76 | 1.76 | 5.14 | 3.14 | 2.28 | 0.28 | 0.259 | **0.645** |
| NEPT603 | 4.07 | **2.07** | 5.27 | **3.27** | 2.73 | 0.73 | **0.416** | 0.509 |

