# Peer review of "Biological and environmental rhythms in (dark) deep-sea 1 hydrothermal ecosystems 2"

_Biogeosciences, 2016_

## Short Comment (SC1) · 16 Dec 2016

This is an interesting study in which the authors report on early results from deep-sea observatories. As the importance of the deep sea and the significance of surface-to-bottom connections become more widely recognized, there is great value in the descriptions in this paper and the techniques being used to explore the role of environmental variables.

In particular, the temperature data presented in this paper contradicts the widely-held view that the deep sea is a stable and 'timeless' environment. It is really interesting that at the NEP, temperatures showed tidal periodicity. This high-frequency variability of temperatures, if related to feeding behavior or chemosynthesis, enables analogy to

the diel cycling of photic ecosystems. Furthermore, the data presented in this paper on tubeworm visibility (presumably a proxy for feeding), suggests a relationship between temperature and activity at NEP. The longer-term (9-month) data on temperature also suggests trends on sub-annual to annual timescales, providing another timescale of deep-sea variability. These findings on temperature open more questions as to the role of surface processes (seasonal cycles, climate modes, deep water formation, anthropogenic influence) in determining the assembly and functioning of deep-sea communities.

Overall, this paper is very results-focused, with a limited discussion of ecological implications. Most significantly, there are two features of the data that are interesting to me, but which were not explored.

The first is the temporal resolution of the data. Given the 6-hour resolution of the imaging data, I find the recurring importance of 6-hour and 12-hour periodicities and lags to be a bit disconcerting. It makes sense to treat the two datasets identically, defaulting to the one with lower temporal resolution. However, the NEP site has continuous images available, and these could be analyzed at a finer temporal resolution. It would be nice to see some discussion of the effects this might have on the results. Analyzing the NEP data at higher temporal resolution would also enable an analysis to see if the 6-hour 'tidal' periodicity is robust. It would also be helpful to more precisely discuss the link between geographic location and the tidal lag; it is not immediately obvious to me that the two locations should show that kind of synchronicity in tidal cycles, because they are in different basins and their latitudinal positions vary by 10 degrees. On this topic, I also wonder why the authors chose to analyze the percent coverage of microbial mats on a 12-hour frequency. (Presumably this is done in some kind of image-processing software and so should require far less effort than the other analyses that are done on a 6-hour basis.)

The second feature of the data that I believe should be addressed is the difference in the size of the areas studied. There is approximately an order of magnitude difference

in the surfaces analyzed; but, this difference is never explicitly mentioned in the text or in the caption of the schematic figure. Looking at the two schematics (Fig 2), if the MAR schematic were divided into 10 pieces, I predict that there would be significant variation among those pieces in terms of taxonomic densities. This fine-scale spatial variation seems important to the ecological conclusions that are drawn, especially since there was far more periodicity seen at NEP (smaller FOV) than at MAR (larger FOV). If these taxa (particularly the polynoids and buccinids) move (whether for foraging or other reasons) over a space that is larger than the NEP FOV but smaller than that of the MAR FOV, this could account for their periodicity in one site and not the other. Given the depth and temperature of these sites, and the taxa in question, we might expect slow movement. Therefore, the 6-hour or 12-hour periodicity could merely be an artifact of the time these animals require to cover their foraging range.

Another area where the paper would benefit from an expanded discussion are the multivariate regression trees (MRT) and redundancy analysis (RDA), which led to temporal split groups and an ordination plot, respectively. What do the temporally consistent groups tell us about the functioning or succession of this ecosystem? Do we expect this community to exhibit variation in densities of various taxa on a timescale of 23 days or less? Since few of these taxa are expected to have trophic interactions between them, are they likely to compete for space, for nutrients, for uptake of microbial biomass? Given that temperature is the only environmental variable included in the RDA for NEP, is it surprising that RDA1 explains so much of the variance? What additional information about the vent communities can we gain from Figure 8?

Finally, I have a few other small comments:

1. Figure 2: I am unable to find the white arrow that is referenced in the caption. Is it blending into the white used to indicate the probe? These two schematics would also benefit from having scale bars, or alternately you could include the area represented by them in the caption.

2. Figure 8: Would it be possible to adjust the axis limits? RDA1 (horizontal position) should be much more important than RDA2 (vertical position), but this seems to be obscured by the way that the data are displayed. Also, there is a discrepancy between the figure and the text for the NEP RDA1 % variance explained.

3. Section 4.1.3, on the regional taxa, is purely descriptive and for that reason would fit better in the results rather than the discussion.

---

## Referee Comment (RC1) · Anonymous Referee #1 · 22 Dec 2016

General comments:

The paper by Cuvelier et al. is an interesting study that uses time series analyses, conducted concurrently at two different hydrothermal vent settings in two different oceans. It is a unique study that deserves attention and it is good to see such work being done. However, there are some important scientific issues that need to be addressed.

A major finding of the paper is that patterns in temperature and tubeworm behavior were seen at both the Pacific (NEP) and Atlantic (MAR) sites that correspond to 6 hour time intervals, which the authors conclude is linked to tidal patterns. Additionally, they note that the same effect is seen 6 hours apart between the two sites which is a product of the time difference between the two sites. The 6 hour periodicity might be present,

however, the link to tidal patterns is not sufficiently developed. There is no data on the tidal rhythms or whether the increases or decreases in tubeworm appearances or temperature values correspond to specific events of the local tidal patterns. In order to come to the conclusion that the periodicity seen in this study is indeed linked to the tides, tidal data needs to be examined and presented within the context of the results of this study.

The other major issue I have with the manuscript in its current form is the use of statistical tests. Some of them are not quite appropriate and others can be tweaked. Details on this are listed below, under specific comments.

It appears from the results, that by and large, not a lot of changes overall were seen. The mussel and shrimp densities at MAR and the pycnogonid densities are the only ones that show an increase over time. This brings up a number of issues and considerations that ought to be treated in the discussion of the paper. For example, one major issue is the spatial extent: the areas analyzed are very small and the authors should include a discussion of the spatial scales at which appreciable changes in the megafaunal community can be observed. In the cases of the increases in densities of taxa, it is surprising that the discussion includes no references to successional patterns. The authors do mention that the mussels represent a climax community at shallow Atlantic vent sites, but there is no discussion of recruitment or colonization as being possible explanations for the observed increases in densities. And, the overall stability is not discussed very well either. Though there is a brief reference to differences in the level of dynamism in vent communities being possibly linked to spreading rates, this is not discussed very much despite stability being one of the major findings.

The writing itself needs considerable improvement. First, it should be read by a native English speaker since there are a number of grammatical errors and sentences that appear to be lost in translation. Secondly, the discussion, particularly the part with reference to the different taxa is written as a list of short, highly abbreviated paragraphs. This needs to be improved upon, restructured and rewritten so that a cohesive story

is presented as opposed to a list of short comments. For example, paragraphs should not end with a new thought or idea such as line 432, on page 12 which states 'Both species were considered predators or scavengers.' This is an important aspect to the biology of the snails discussed within this paragraph, without a doubt, but it is something that should be expanded upon, and should not be the final, concluding sentence of a paragraph that up to that point has not made any mention of trophic relationships or feeding biology. As it stands now, this part of the discussion reads basically like bullet points instead of a cohesive discussion.

Specific comments:

Introduction: In the key questions in the last paragraph: the first question is 'are tidal rhythms discernible in both vent settings?' It would be better to perhaps say 'are rhythms discernible in both vent settings that correspond to tidal patterns?' Since making the actual connection between the patterns seen in this study and tides is beyond the scope of the study.

The introduction should include some background about the major faunal groups and community structure at the two study sites. This is presented currently in the Methods section and certainly more details can be presented there, but the Introduction should also contain this information because understanding the settings is important contextual information.

Methods and Results: A number of key methodological information is missing. Though it is mentioned that the MAR observatory was positioned to face the Eiffel Tower edifice, no such information is given about the NEP observatory, such as whether it is also facing a chimney structure or not. If it is also placed facing a chimney structure, then this should be clearly stated early on in the manuscript, because chimney communities differ from areas of diffuse flow (and even host different morphotypes of Ridgeia tubeworms) which would mean that this study is examining chimneys on vents from two different oceans, which is very specific.

It is not mentioned, but clear from the photos, that the camera is positioned facing forward. In this case, there has to be clear details on how the spatial extent of the field of view was calculated. This is very important information and I am surprised that it has been left out. Other details about the imagery is also missing, for example, since video cameras were used, I assume that video stills were taken at the appropriate time points and those video stills were analyzed and used for marking the animals (in which software?), but these details are not present in the manuscript.

I think that it is inappropriate to use tubeworm abundances or tubeworm densities since in reality, what was counted where the extended plumes. Throughout the text, this should be changed to visible plumes or extended plumes, etc. and not tubeworm density.

In general, density should not be used at all. In both cases, the surface filmed and analyzed is considerably less than 1 m2 which means that all the density numbers are extrapolations and I don't think that is appropriate. Since within a site, the same area is filmed and examined for all 23 days and time points, the use of numbers of individuals instead of extrapolated densities would be more appropriate. Similarly, for microbial mats, use area coverage instead of percentage of area (and was percentage and density calculated based on filmed area or analyzed area?)

There is no explanation as to why areas of microbial mats were examined at 12 hour intervals and not at 6 hour intervals like the fauna.

Due to the difference in depths and ambient temperatures between the two study sites, raw temperatures should not be used at all. Instead, rescaled temperatures (raw temperature – ambient temperature) should be used and presented. The authors even say that there is a 2 degree difference in ambient temperatures between the sites and they say that even when this is taken into account, the NEP temperature recordings have a higher mean and maximum temperature. However, that does not mean that the distributions are necessarily different. A simple t test should be done to test if they

are significantly different or not. The temperature data shown, for example, in Figure 5 seems to indicate that they are not significantly different since they appear to basically differ by about 2 degrees, which is the difference in ambient temperature between the two sites.

I am not convinced it is appropriate to use a linear regression model to state if changes in densities over the 23 day period were significant or not. The independent variable is time, which is actually specific time points. It is important to have Figure 3 to show the trends, but fitting a line to these data and using that to say the changes are significant or not is, I believe, incorrect. The buccinid density graph really illustrates this, where the densities increased, then decreased and then increased again. That clearly does not mean that overall, in the study time period, buccinid densities showed a decrease, or should be represented by a downward sloping best fit line (as it is in the paper).

The differences in analyzed areas between the two study sites needs to be considered very carefully. I understand that the setup could not accomplish getting the same spatial extent for the fields of view, certainly, that would have been near impossible to achieve. However, when comparisons are made, for example, in the discussion about pycnogonid densities differing greatly between the two study sites, this difference in FOV extents needs to be kept in mind. In fact, it would be very difficult to constrain whether differences in densities or numbers of a specific taxon between the two study sites is a real difference or due to sampling artifacts. Therefore, such discussions need to be treated very cautiously.

There are some inconsistencies in terms of what was analyzed. For example, anemones are mentioned in the text, but are not in Table 2 which lists all the animals analyzed. Similarly, in the results (lines 229), mention is made of ophiuroids, which are not mentioned anywhere else before. And line 232 talks about a fish, which is also mentioned in Table 2, but was actually not seen in the stills, but in other video footage, which means, it was seen at other time points. Discussion of trends seen outside the time points relevant to this study should be discussed separately because it is has the
potential to introduce bias (large, flashy fauna are easily seen and focused on).

Limpets are mentioned and it is also said that they were not quantified (understandably so, because they are very small and numerous), but they are not shown in Figure 2.

In general, the results and the discussion appear to have three major themes that should be dealt with in separate sections. The first is spatial trends and associations between taxa within each study site, the second is comparisons between the two sites and the third is temporal trends. These are often intermixed and the paper would benefit by having them discussed separately. There will be some overlap between them, but currently, the results and discussion comes off as being very patchy and leaping from one point to another, without complete development of each point. Splitting into different sections might help to make the paper more cohesive.

I suggest adding two figures or analyses: first, in addition to figure 3, which shows densities plotted for the different time points, the authors could benefit by having a similar figure, but with difference in numbers from the previous time point (6 hours) on the x axis instead of numbers.

Secondly, I strongly suggest having a figure with tubeworm appearances (and anything else that shows the 6 hour pattern) vs. temperature. And in fact, regression models could be applied to these and it would strengthen your case that temperature can be used to predict tubeworm behavior.

The discussion about the same taxon inhabiting bare substrate at one site but not at the other is very problematic, because the FOV for NEP does not include bare substrate at all. In fact, the caption for Figure 2 even lists bare substrate as being an MAR only feature. If bare substrate is not present in the images of NEP, then it is not possible to say that NEP taxa that are seen on bare substrate at MAR are not seen on bare substrate at NEP.

In the discussion, certain taxa names are introduced for the first time, e.g., Bythograeidae, Bythitidae, and Majidae. These names do not appear in the Introduction or Methods, even when the animals are being introduced and they do not appear in Table 2 which lists the animals studied. The manuscript would benefit by keeping reference names for taxa consistent throughout the manuscript.

The first part of section 4.2, ie, the discussion about mussel valve openings is problematic. By opening valves, do the authors mean that one of the siphons are visible and extended or simply open? Mussels filter water through their inherent and exhalant siphons and fully opened valves are generally only seen in sick or dead individuals. Therefore simply talking about mussels valve openings does not seem appropriate, or should be explained further.

I do not know what software was used to mark and count the animals, but if the animals were physically marked, then it might be a good idea to examine the extended tubeworms more closely to see if there is periodicity in appearances among individuals. For examples, are half the worms extending out of their plumes at a certain time while the other half remain in their tubes and at the next interval, do you see the retracted ones extended and the extended ones retracted, or is it random in who is retracted or extended at any time point?

When talking about periodicity of the more mobile animals like pyconoginids and snails, etc., it is important to keep in mind the time and spatial scales: Currently, I don't think it has been shown conclusively that the observed periodicity is real periodicity and not the result of mobile animals moving in and out of a small area of focus at their own individual paces.

Though there is information on and a discussion of the CHEMINI system for measuring iron, no discussion or mention is made of sulfide. This is a very big gap in the discussion since sulfide is the fuel for the chemosynthesis based animals, and also a determinant of other animal distributions due to its toxicity. I understand that there was so sulfide sensor and therefore real sulfide measurements were not possible. However,

temperature, oxygen and iron are correlated with sulfide and can be used as a proxy to a certain extent for sulfide. Even if real concentrations of sulfide are not included, sulfide itself should be discussed because it is the source of energy in this system and one of the main reasons why tubeworms extend out of their tubes.

Technical corrections:

Please proofread for corrections to English grammar and sentence constructions.

Figure 1: The inset pictures are very small, and I think, the ones showing the FOVs are not necessary here, since they are presented in Figure 2. A better figure would be the map and the instrumentation. If the authors do decide to include the pictures of the FOVs, please make sure that the caption states clearly what all the images are. Currently, the caption does not explain what the smaller pictures are.

Figure 2: In addition to the sketches with the animals and substrates interpreted, one sample image in its original form, without interpretations drawn in, needs to be included as well for each site. Ideally, instead of a composite sketch, just one sample image should be presented, with and without the interpretations drawn in (and a reference can be made to Table 2 for a comprehensive list of animals seen at the two sites). This provides the opportunity to see what is being analyzed. These images also need scale bars. And, the white arrow that is mentioned in the caption, which is supposed to be pointing to the fluid exit, is not in the figure. Additionally, there is no mention whatsoever, of 'mussel background' anywhere in the text but it is drawn in in this figure.

Figure 4: The x axis is labeled incorrectly on the figure: it states 'hours', but the scale bar reads 0 to 40, but it should read 0 to 552 if it is hours. The caption reads that the x axis contains periods of 12 hours and this makes more sense, since 552 hours would equal to 46 12 hour periods. Secondly, as mentioned before, real areas should be used instead of percent areas. In fact, this is a reason why using percent cover is inappropriate: since the MAR FOV is much larger than the NEP FOV, the use of percent cover gives a very different view, namely that much more of the NEP is covered

in microbial mats than at MAR. This is not necessarily true, it just so happens that the area in question at the NEP site is much smaller and a similarly sized microbial mat there gives the impression of being much larger because the overall study area is much smaller.

Figure 7: what are the dashed lines?

Figure 8: The caption should mention why there is a box drawn in the graph for NEP Fe.

Table 1: remove coordinates and write out the full form of latitude and longitude. The last line, for turbidity has a '/' for NEP, this should be changed to N/A.

Table 2: In number of images, please spell out that 93 is the total, and 9 or 5 are the number of images that are missing, or could not be recorded. However, given that in both cases, video stills were taken, is it not possible to take an image just before or just after the specific time in question?

For surfaces, perhaps cms might be more appropriate since they are both much smaller than 1 m2.

Surface analyzed: it says to refer to Fig X, please change to refer to the correct figure in question.

The listing of taxa in this table needs to be more consistent. For example, if you put a descriptive category in the left column ('engineering species') then similar descriptive terms should be used for the others (mobile predators, scavengers, etc.). Basically, the same general type of information should be in the same column, instead of having a descriptor in one row and class or phylum names in the others. In the second and third columns, the order should be consistent. For example, you start with phylum (Mollusca), then family (Mytilidae), followed by common name in parentheses and the next line has the species name, which is a good format to follow. Similarly, for NEP, it should then read Annelida, Siboglinidae (tubeworms) and the species name on the next

line. So, next, should be Annelida, Polynoidae (scaleworms) and then multiple species on the next line. With M. fortunata, these higher categories and common names are left out (and / should not be used to indicate not available). Finally, since anemones are also present and discussed, they should be included in this table as well.

Table 4: As mentioned before, conduct a statistical test on the distributions of the rescaled temperature values to see if they are significantly different or not and include the results in this table.

---

## Referee Comment (RC2) · Anonymous Referee #2 · 28 Dec 2016

GENERAL COMMENTS

This paper is a fine example of new scientific inquiry enabled by deep-sea observatories. Performing paired high-resolution time-series studies is a particularly novel and interesting aspect of the study. Although the technology and methods used are still relatively new (and exciting), I found the authors neglected discussing vent ecology/animal physiology (i.e. mechanisms driving the patterns) to focus on methods and data collected. The data aligns with the scope of BG, but the text requires work addressing specific interactions.

This paper is an important stepping stone to better understanding the deep-sea hydrothermal vent environments. Although the findings are not exactly conclusive, there

is valuable information presented here, about the tools and apparent (and lack of apparent) environmental and ecological temporal patterns.

In general, I found the manuscript was well written, and the language used to be fluent and precise. That said, inconsistencies in formatting were very evident –this was distracting and, at times, outright confusing. The figures and tables also require work.

SPECIFIC COMMENTS

Title and abstract. I found both to be slightly misleading. The majority of the study results yielded no evidence of rhythms. This lack of evidence is still a result and it warrants discussion (e.g. Why aren't the majority of vent animals influenced by tidal rhythms?).

L. 55. "...exact same time span and resolution, have been analysed." Not sure I would say "exact": with the differences in gaps, sizes of images, and data collection durations (at times, continuous vs. punctuated). The first two paragraphs of section "2.3.1. Imagery analysis" are to the contrary.

L. 88. Was the lighting different for the different sites? Were the lights on for different durations? Discuss the effect of any variability in artificial light at the sites.

L. 122. Add text about the analysis of microbial mats and the anhydrite (in Fig. 2). Is there any mineralogical work to support the identification of anhydrite (could it have been sulphur precipitate)? How was the white encrusting mineral ("anhydrite") resolved to be different from the white encrusting bacterial mats?

L. 124. Explain the "gaps". Why are there gaps in the data?

L. 127. What was the resolution of the images from the different sites (sub-centimeter)? Were the resolutions actually comparable? Were the cameras/image sizes/distances from substrate the same?

L. 132. "Sketch" suggests artistic, may be better to refer to it as a "map" (i.e. it is

a single photo with overlays representing max. occurrences...). How was this map created? Add information regarding the program and method used.

L. 138. Does "Fig. 1" show this? This figure and its caption don't indicate as much.

L. 223-225. and Fig. S1. Confusing. Consider removing at least the "days" from the text? As it reads now, the sentence references a Fig. with an x-axis in periods (which equal 6 hours), 18 hr periods, hours, days, and hours in multiples of 18. This is too much. Also, consider changing "*" to "x".

L. 247. I don't see how Fig. 2 demonstrates this point: it's a 2D schematic with no information about the substrate below the mobile fauna.

L. 399. Add a sentence describing the diversities.

L. 423. The assumption is the same individual is returning every time? Can you really say this?

L. 425. And so?

L. 435. What is the "very distinct spatial distribution in NEP"?

L. 442-443. Unclear what the authors are saying here.

L. 487. How fast do mussels move? Did you expect to see a difference at a frequency of 6 hrs?

L. 594. At vents or everywhere?

L. 607. Review Lau back-arc basin hydrothermal vent studies linking faunal variations with environmental gradients.

Discussion and Conclusion: Explicitly offer at least one mechanism to connect the influence of the tides and temperature, and the influence of tides and the pattern observed in tubeworm appearance.

Discussion and Conclusion: Do the authors believe the tides change the overall temperature of a vent, or just the outflow directionality of the fluid at the point location of the probe?

Discussion and Conclusion: Were all the tubeworms alive? If not, what effect could this have had on the ecological patterns observed/not observed?

TECHNICAL COMMENTS

L. 56-60. Rewrite "Key questions" sections so that the sentences are grammatically correct. For example, "...put forward are: (i) are tidal..." and "the most? And finally, (iv) do ...".

At times, I found the writing was too informal for a scientific manuscript. I was not happy with the (repetitive) use of "vs.", "on one hand...on the other", and "and/or", and [L. 422] "...individuals appeared very attached..." The tense of the manuscript jumps around sometimes. For example, L. 134-135.

There are many inconsistencies in the text formatting: *"Hours" was written as "hours", "hr", and "h", with a space or with no space between the number and the shorthand "hr" or "h". This inconsistency was even more confusing because the UTC time was also reported using "h" (again, with either a space or no space between the number and the "h") or UTC was reported with "AM" or with no units. *In-text citations are inconsistently formatted: "et al.," is often missing a comma; both "and" and "" are used for 2 author papers; multi-paper citations were not always listed chronologically [L. 406]; author's initials included [L. 164]; and missing a comma after authors [L. 186] *Values with units are reported with and without spaces. For example, m vs. m. *The shorthand for "Figure" is written with and without punctuation, within the text and the figure captions (i.e. "Fig." and "Fig"). *Section numbers are written with and without "." at the end (in the section titles, as well as when referred to in the text). *"Oxygen" or "oxygen". *Text jumps between "iron" and "Fe" in same paragraph. *Formatting the title of a subsection varied between: title in the text (e.g. L. 253 and title on a separate line (e.g. L. 341); indented or not; followed by a long/short/bolded/no dash. *Mean and

stdev written: $\pm$ units, $\pm$ units, and $\pm$ units. *Within the same paragraph, reporting a date range changes from "date to date" and "date - date". *The Reference section requires some attention. For example, "Year" vs. "(Years)"; ending the authors list with a ","; inconsistent formatting of the volume number, issue number, and page text; inconsistent spacing; inconsistent punctuation; different color text [L. 839-840]?; etc.

L. 165-166. Insert space

L. 177. Remove "()".

L. 198-200. Poorly written. Rewrite sentence.

L. 237. Change "featuring" to "with".

L. 272-273. (as one example)Watch the p-value sig. figs.; at times, they vary within the same sentence. Personal preference: never report p = 0 (or in this example, "0.00"), report it as p < 0.001.

L. 305. Reference Fig. 5 somewhere in the paragraph.

General: Write out values less than 10 (e.g. 9 months –> nine months)

L. 339. Repetitive.

L. 344. Use "...was already.." or "...as well", but not both.

L. 430. Correct. "...abundant on to areas..."

L. 432. Correct. "...both species [are] considered..."

L. 474. "...feeding [activity]..."?

L. 494. Open bracket with no closing bracket.

L. 496. Delete "Until now", because it still has not been established.

L. 508. "...by a [longer] study..."

L. 510. "...as they [become] more..."

L. 522. "...in a single taxon..."

L. 545. "...for both [temperature] probes..."

L. 567. "...were close to..."

L. 572. "...Tunnicliffe et al., 1997)."

L. 614. What is meant by "harshness"?

L. 617. "...and [piloting] skills..."?

L. 629. "This is [likely] due..."?

L. 635. Capitalize "automated". Do they "need" to, or would it be helpful? Suggestion: "faunal abundances [in] images."

Figures (in general): Consider (i) standardizing graph formatting throughout the manuscript, (ii) removing repetitive information in graph titles (e.g., Fig. 6: use the probe name only vs. "T-MAR for imagery duration - hourly average"; that information is in the caption), (iii) clean up the axis ticks, labelling, and titles, and (iv) move footnotes (denoted by an asterisks, "*") at the end of a Fig. caption.

Fig.1. This figure is missing some key information. The text for the scale bars is too small. Why is there text and colour bars in the lower right-hand corner of the NEP bottom inset? Label Canada and/or USA? Label the oceans? In the caption, explain or refer to the 4 insets. What are we looking at here? Consider providing larger photos? Add punctuation for "Fig. 1." and "Matabos et al.".

Fig. 2. Is it necessary to retain some transparency (the key colours really do not match the colours overlaid as semi-transparent)? Change to "Microbial [c]over" (in key). Why is the text "Ifremer" in the bottom right corner and why is it coloured in as "Pycnogonida" (in yellow)? The hatching in the MAR image (for "Mussel background"

is difficult to resolve. Add punctuation for "Fig. 2.". Move footnote to the end of the caption?

Fig. 3. Graphs and text are grainy. Consider deleting "densities" from the 10 individual titles (repetitive), and just list the taxa. Reduce the number of x-axis ticks (I can't tell which line is associated to the values listed). Add y-axis title. The Crab graph is missing the number "10" on the y-axis. Shorten the number format of the y-axis for the MAR Pyncognoid and Shrimp graphs (i.e. 0, 1, 2... vs. 0.0, 1.0, 2.0...). Mention "23 days" in the caption. Change to "...with an "*"" OR "*Taxa with significant trends."

Fig. 4. When printing in black and white, it is impossible to tell the difference between light blue and light gray. In Fig. 5, NEP is black and MAR is light gray (which can be distinguished in black and white print). To standardize the figures, and for printing purposes, consider changing NEP to black and MAR to gray for Fig. 4. Consider rewriting caption and/or changing the x-axis title. I'm not sure what the value is supposed to be, hours or periods? Reorganize so the sentence doesn't start with "

Fig. 5. To save space, consider adding a 2nd axis to the temperature graph (to display both NEP probe temperatures, instead of repeating the MAR data. Remove "short-term" for graph titles? If not, change to "NTU short-[t]erm". Is it necessary to repeat the same key for 3 of the 4 graphs? Although this is not the only time the figures include stacked graphs, this is the only time the x-axis is included.

Fig. 6. Shorten the y-axis labels to represent a count of the days (e.g., day "1", "2"..."23" vs. "2011-10-07", "2011-10-08"...). Add titles for the x- and y-axis (e.g., "day" and "hour"). Indicate somewhere in the figure or caption: temperature in °C.

Fig. 7. More information is required for the caption. Are there gray and black vertical lines (appear to be)? If so, what do they represent? What are the 2 blue dashed horizontal lines on each graph? Change "X-axis" and "Y-axis" to "x-axis" and y-axis (to be consistent with text).
Fig. 8. Label and mention: one graph is MAR and the other is NEP. Change text and vector lines to black (vs. blue). Difficult to read the text on the graph, increase the size? Define RDA? There is a noticeable difference in the size and quality of text in the left and right graphs. Standardize?

Fig. 9. Is the x-axis in hours? Include "Temperature (°C)", not just "°C" for y-axis. Remove redundancy in the graph titles and consider adding this information to the figure captions, e.g. "...over six and nine months".

Fig. 10. Confused again by the x-axis title and the caption. This data is for a one-week period equalling 200 hours, but the x-axis title is "Period", not "Hours", and plus, 1 week = 168 hours. Please clarify. Change the lines to black (no need to be coloured red).

Fig. S1. Include the information for the white vs. black symbols. Why change the x-axis intervals? If each period = 6 hrs, and there are 45 periods, the graph represents 270 hrs or 11.25 days. Include this easy to understand temporal reference (and why this length of time)? Change the lines to black (no need to be coloured red).

Fig. S2. Similar concerns to Fig. S1. Why change the x-axis intervals? Remove the repetition of the x-axis title (i.e. only include "Period" once). Change the lines to black (no need to be coloured red).

Fig. S3. Change to "...(a) MAR and (b) NEP...". Are the "random" data consecutive? Why not report the specific month and year (even if it was selected randomly)? Use the same style quotation marks at start and end of the quote -or in this case, consider not using quotation marks at all. Many of the same comments and concerns as expressed for Fig. 6.

Tables (in general): Consider (i) reducing the number of lines (vertical and horizontal) for each table, (ii) removing repetitive information (e.g. "2011-2012"), (iii) condensing the area of each table (there is often a lot of blank space between rows), (iv) use either "Table :" or "Table .", but be consistent, and (iv) move footnotes (denoted by an

asterisks, "*") below a Table.

Table 1. "[o]xygen". "[T]wice". Use "NA" instead of "/" (or define "/"). Be consistent, "min" or "min.". If minutes = "min", seconds could = "sec". As in the text, include "at" when listing the sample times (e.g., at 2h, 6h...UTC). Be consistent with apostrophe symbols for the coordinates (styles change between MAR and NEP). Explain/include row title for "Wireless" and "Cabled".

Table 2. Move footnote to below table (or at least the end of the caption). "[A]re visiting...". Fix: "see fig. X"? Reverse how the gap range is reported ("9 to 93 gaps" and "5 to 93 gaps")?For surface filmed and surface analyzed, be consistent with sig. figs. and with the information provided (why list the ca. dimensions 2 out of 4 times?). Reported frequency as "6hr" and "12 h" in the same table (use a consistent format). Check citation formatting (missing punctuation). Use "NA" instead of "/" (or define "/"). The lines of this table are bolded, why?

Table 3. n = ? (photos?) Missing "h" after "553" twice. Add a space to "( 2 days)".

Table 4. Include "°C" in the table caption and remove it from each record. "[S]tdev"?

---

## Author Comment (AC1) · 14 Feb 2017

Short comment - Christina Hernandez

We would like to thank Christina Hernandez for her point of view and comments on the manuscript. We addressed questions raised below.

Overall, this paper is very results-focused, with a limited discussion of ecological implications. Most significantly, there are two features of the data that are interesting to me, but which were not explored.

The first is the temporal resolution of the data. Given the 6-hour resolution of the imaging data, I find the recurring importance of 6-hour and 12-hour periodicities and lags to be a bit disconcerting. It makes sense to treat the two datasets identically, defaulting to the one with lower temporal resolution. However, the NEP site has continuous images available, and these could be analyzed at a finer temporal resolution. It would be nice to see some discussion of the effects this might have on the results. Analyzing the NEP data at higher temporal resolution would also enable an analysis to see if the 6-hour 'tidal' periodicity is robust.

High resolution data for the NEP for the same period was analysed and published in Cuvelier et al. 2014 PlosONE. That paper also explains why a one-hour resolution was used for imagery assessment as well as the limits of the duration. While longer time-series are available for NEP and in mean time for MAR as well, due to new, more recent deployments; these fell out of the scope of the current article, but they are part of ongoing research projects.

When comparing two datasets, collected separately, the data is inherently limited by the lowest resolution and shortest time span.

The scope of the current study was comparing community dynamics at both sites, which represents a unique dataset at a 6h frequency. 6h periodicities were only revealed in temperature when analysed on an hourly frequency, on short and longer time-scales. Impacts of duration and resolution were addressed in Cuvelier et al 2014 PlosONE and were referred to in the presented manuscript when deemed necessary. The differences in periodicities revealed are due to the frequency analysed and/or the sampling interval. Comparison to Cuvelier et al 2014 PlosONE serve precisely to discuss and draw attention to the importance of recording frequencies both in imagery and environmental variables, which resulted in one of the main conclusions (e.g. L517-520, L632-633).

It would also be helpful to more precisely discuss the link between geographic location and the tidal lag; it is not immediately obvious to me that the two locations should show that kind of synchronicity in tidal cycles, because they are in different basins and their latitudinal positions vary by 10 degrees.

It is exactly the correspondence between the time difference and the lag that makes it such an interesting observation. To our knowledge no such observation in the deep sea was done before; hence the interest in publishing it, even though it might raise more questions than answers at this point. More significant lags were revealed (see Fig. 7.), but those that were most significant ranged between +4 and +7 with tapering occurring in both directions from the 5-6h peak. The 6 hour difference corresponds to the difference of 90 degrees in their longitudinal position. We do not know at this stage if it is (solely) related to the geographic locality, or if, among other possible factors, the difference in depth is at play as well (1700m vs 2200m).

On this topic, I also wonder why the authors chose to analyze the percent coverage of microbial mats on a 12-hour frequency. (Presumably this is done in some kind of image-processing software and so should require far less effort than the other analyses that are done on a 6-hour basis.)

Microbial mat coverage, while indeed taking less effort than assessing densities, was presented based on a 12-hour frequency, because there was not much variation at a higher resolution and thus would not contribute much to the discussion at hand.

The second feature of the data that I believe should be addressed is the difference in the size of the areas studied. There is approximately an order of magnitude difference in the surfaces analyzed; but, this difference is never explicitly mentioned in the text or in the caption of the schematic figure. Looking at the two schematics (Fig 2), if the MAR schematic were divided into 10 pieces, I predict that there would be significant variation among those pieces in terms of taxonomic densities. This fine-scale spatial variation seems important to the ecological conclusions that are drawn, especially since there was far more periodicity seen at NEP (smaller FOV) than at MAR (larger FOV). If these taxa (particularly the polynoids and buccinids) move (whether for foraging or other reasons) over a space that is larger than the NEP FOV but smaller than that of the MAR FOV, this could account for their periodicity in one site and not the other. Given the depth and temperature of these sites, and the taxa in question, we might expect slow movement. Therefore, the 6-hour or 12-hour periodicity could merely be an artifact of the time these animals require to cover their foraging range.

The differences in surfaces analysed were stated in L127-129, L198-200, Table 2 and its legend. Densities were used to counteract the effect of the size of the FOV, without aiming to extrapolate densities and observed variations for the entire edifice.

Significant faunal periodicities were observed on both sites for polynoids and on the NEP for buccinids and tubeworms. While not observed in this study, we now have evidence that mussel valve openings also follow a tidal rhythm on the MAR site (Matabos et al. unpublished data) (L497-498). Differences in the FOV and even more the 3D physical structure of the two assemblages could obfuscate rhythms in faunal abundance or behaviour.  If a periodicity is observed on a small FOV, it is more than likely that it will be observed on a larger FOV as well. The presence of tidal rhythms in the organisms are thus likely to apply for the entire hydrothermal structure. The results obtained here are nevertheless linked to the size of the FOV, limiting interpretation at larger scales without additional information (imagery at larger scales and high time resolution recordings).

By foraging range, we assume you mean spatial foraging range. The presence of tidal periodicities in mobile organisms within the FOV would be rather due to changing local environmental conditions (thus reflecting the tidal periodicities herein) which influences them to move into certain areas where and when conditions are favourable, e.g. when a region is temporary not exposed to fluid flow due to tidal currents. However, it is nearly impossible to know if mobile individuals entering the FOV are recurring visitors or not (only when linked with possible homing behaviour it could be assessed, but see below) which makes the assessment of a possible spatial foraging range following a certain periodicity very difficult. Nevertheless, the presence of a tidal periodicity in faunal densities as observed in our study seems unlikely to be due to a "spatial foraging range" artefact for a number of reasons:

(1) Strongest periodicities were revealed in sessile animals (tubeworms) and was most likely a reflection of their feeding needs linked to the presence of hydrothermal fluid (see L495-497).

(2) The tidal periodicities observed in the fauna are due to the environmental variables such as fluid flux, for which temperature is a proxy and which shows tidal periodicities (L41-42).

(3) Polynoid and gastropod taxa at both study sites are different and appear to have different microhabitats and niches. Their periodicities remained indicative, i.e.no or little significant periods. Their grazing, scavenging or predatory behaviour could be responsible for them to move out of the FOV because their food source has been depleted. Or they could move in or out due to locally (un)favourable environmental conditions, which would be food-related for certain organisms that rely solely on chemosynthesis through symbiosis, though not for active feeders. Even with the possible homing behaviour observed in one large Polynoid individual on the MAR, no periodicities could be revealed based on its presence or absence.

Another area where the paper would benefit from an expanded discussion are the multivariate regression trees (MRT) and redundancy analysis (RDA), which led to temporal split groups and an ordination plot, respectively. What do the temporally consistent groups tell us about the functioning or succession of this ecosystem? Do we expect this community to exhibit variation in densities of various taxa on a timescale of 23 days or less? Since few of these taxa are expected to have trophic interactions between them, are they likely to compete for space, for nutrients, for uptake of microbial biomass? Given that temperature is the only environmental variable included in the RDA for NEP, is it surprising that RDA1 explains so much of the variance? What additional information about the vent communities can we gain from Figure 8?

23 days was the limit of the time-series as explained in L117-120 and appears too short to reveal any apparent succession patterns. Intra-annual variations in hydrothermal vent communities have been suggested to be at play but have only seldom been observed (Tunnicliffe 1990, Copley et al 1999). At what time-scales exactly we can expect to observe changes in communities or successional patterns is unknown. Time scales of succession are dependent on location and spreading rate with decadal scale stability observed e.g. on the Mid-Atlantic Ridge. However, even though decadal or supra-annual stability at the scale of entire edifices can be observed, it is important to keep in mind that changes on shorter time spans and smaller scales do occur (e.g. Cuvelier et al 2011 L&O). This study contributes to this caveat in our knowledge by revealing possible small time-scales at play, among others through the MRT's.

Mobile, semi-sessile and sessile organisms were quantified within the FOV of the observatories, so changes in densities are to be expected when analysing imagery recorded several hours apart. How they fit in the succession stages depends on the time-scale on which they occur and is restricted by the duration of the time series, which at this stage might prove too short to reveal significant or conclusive results. How exactly they impact the other assemblages and the entire edifice community remains to be determined and for this purpose larger scale observations or multiple deployments on different assemblages representing various stages of succession are needed.

Why would we assume these taxa to have few trophic interactions? After all they are dependent in first or second degree of the hydrothermal fluid flow and associated microbial composition which inherently lead to competition and other interspecific interactions and a complex food web.

About RDA1 explaining most of the variance: RDA1 indeed tends to explain higher proportion of variance than RDA2, 3 etc. The high proportion of variance explained here by RDA1 is attributed to the temperature for both MAR and NEP RDA's even though the MAR RDA incorporated one additional environmental variable (NTU). These ordination plots thus corroborate the conclusion in L610-611 that temperature, to date, appears to remain the best proxy for hydrothermal vent community dynamics even though not all community dynamics can be explained by it. When more environmental variables are incorporated in the RDA's this will impact the proportion of variance explained by each axis, which is also visible in Fig 8. RDA's were carried out for the sites separately,

i.e. MAR and NEP at different frequencies to include all local environmental variables available (not shown) and revealed no significant influence of these environmental variables, in other words, temperature remained the most explanatory. Due to the differences in recording frequencies (e.g. Fe) and low reliability of the oxygen measurements, only temperature and NTU were withheld which also allowed comparison between the 2 sites.

Finally, I have a few other small comments:

1. Figure 2: I am unable to find the white arrow that is referenced in the caption. Is it blending into the white used to indicate the probe? These two schematics would also benefit from having scale bars, or alternately you could include the area represented by them in the caption.

The arrow disappeared underneath the layers but will be put to the foreground again.

Scale bars were present on the sample images present in fig. 1 and areas represented by both areas are mentioned in table 2. Based on the reviewer's comments sample images with scale bar originally presented in fig. 1 have been added to fig. 2.

2. Figure 8: Would it be possible to adjust the axis limits? RDA1 (horizontal position) should be much more important than RDA2 (vertical position), but this seems to be obscured by the way that the data are displayed. Also, there is a discrepancy between the figure and the text for the NEP RDA1 % variance explained.

There is no use to rescale the axes because it is a representation of the ordination plot and the variation explained by each axis is mentioned thus allowing for a correct interpretation. Different weights could be attributed to the plotting which would have a repercussion on the representation of the axes, however, they do not enhance interpretability nor readability of the plot.

The discrepancy between text and figure is taken care of.

3. Section 4.1.3, on the regional taxa, is purely descriptive and for that reason would fit better in the results rather than the discussion.

We prefer to leave it in the discussion section since it links the results in faunal densities/presence/absence with existing knowledge and literature on the taxa.

---

## Author Comment (AC2) · 14 Feb 2017

Referee 2

We would like to thank the reviewer for a thorough revision of the submitted manuscript. We took into account all issues raised and addressed them below. Additional more editorial comments on tables, figures and overall word choice have improved overall consistency of the submitted manuscript.

Although the technology and methods used are still relatively new (and exciting), I found the authors neglected discussing vent ecology/animal physiology (i.e. mechanisms driving the patterns) to focus on methods and data collected. The data aligns with the scope of BG, but the text requires work addressing specific interactions. This paper is an important stepping stone to better understanding the deep-sea hydrothermal vent environments. Although the findings are not exactly conclusive, there is valuable information presented here, about the tools and apparent (and lack of apparent) environmental and ecological temporal patterns. In general, I found the manuscript was well written, and the language used to be fluent and precise. That said, inconsistencies in formatting were very evident –this was distracting and, at times, outright confusing. The figures and tables also require work.

Valorising this reviewer's comment, as well as taking reviewer 1's comments into account, paragraphs in sections 4.1 and 4.2 are reorganised, though main lay-out is withheld, and more relevant information on the animals discussed was added. Links with their physiology are included when linked with environment, e.g. on tubeworms (see reply below). Overall, less significant interactions were observed than revealed by higher frequency analyses (e.g. Cuvelier et al. 2014, PlosOne).

SPECIFIC COMMENTS

Title and abstract. I found both to be slightly misleading. The majority of the study results yielded no evidence of rhythms. This lack of evidence is still a result and it warrants discussion (e.g. Why aren't the majority of vent animals influenced by tidal rhythms?).

In our experience, it is rather the opposite: the fact that tidal rhythms are present at deep-sea sites tends to surprise people, hence the more descriptive title. We decided to keep the same title as we do observe rhythms in both biotic and abiotic factors. More specifically, rhythms were found in one taxon at MAR (polynoids) and two (tubeworms and buccinids) at NEP (see L222-224 and L243-246 respectively). Rhythms in temperature were revealed at both sites L348-363).

L. 55. ". . .exact same time span and resolution, have been analysed." Not sure I would say "exact": with the differences in gaps, sizes of images, and data collection durations (at times, continuous vs. punctuated). The first two paragraphs of section "2.3.1. Imagery analysis" are to the contrary.

The word exact was removed.

L. 88. Was the lighting different for the different sites? Were the lights on for different durations? Discuss the effect of any variability in artificial light at the sites.

Lights were on continuously in the period analysed for the NEP (see Table 1, 23 days), contrastingly at the MAR where lights powered on with the same frequency as imagery recording (every 6 hours). This was added in section 2.2. and briefly touched upon in the discussion.

L. 122. Add text about the analysis of microbial mats and the anhydrite (in Fig. 2). Is there any mineralogical work to support the identification of anhydrite (could it have been sulphur precipitate)? How was the white encrusting mineral ("anhydrite") resolved to be different from the white encrusting bacterial mats?

Movement in bacterial filaments allowed to distinguish between microbial mats and encrusted minerals. It is very likely that there are encrusted bacterial mats within the "anhydrite" patch, though due to the colour similarity these were impossible to differentiate and quantify. Unfortunately, we do not have mineralogical work to support the identification of anhydrite. Therefore, we changed anhydrite in the legend of Fig. 2 to "white substratum" and added "possibly anhydrite with encrusted microbial mats" in the legend text.

L. 124. Explain the "gaps". Why are there gaps in the data?

Definition of gap has been added: "The gaps in the recordings were failed recordings (due to observatory black-out or instrument failure) or unusable video sequences (empty, black or unfocused)."

L. 127. What was the resolution of the images from the different sites (sub-centimeter)? Were the resolutions actually comparable? Were the cameras/image sizes/distances from substrate the same?

Cameras used in both ecological modules in 2011 was Axis Q1755.

Distances to the assemblage filmed tend to differ due to module location and proximity to the hydrothermal faunal assemblage and surface size of imagery recorded is thus different. Size of imagery recorded differed slightly, which was reflected in the size of the screen stills taken from the video sequences, which were 1920x1080 pixels for NEP and 1440x1080 pixels for MAR. However, surface filmed differs from the surface analysed see Table 2 (surface filmed: ~0.3802 m² for MAR, ~0.0661 m² for NEP; surface analysed: ~0.322 m² for MAR and ~0.0355 m² for NEP) and Fig. 2.

L. 132. "Sketch" suggests artistic, may be better to refer to it as a "map" (i.e. it is a single photo with overlays representing max. occurrences...). How was this map created? Add information regarding the program and method used.

Sketch has been changed to maps both in text and legends. These maps and the overlays were created in Photoshop, the merge of all images was done with ImgLEP programme (publication in prep), a software developed at Ifremer for (semi-)automated image analyses.

L. 138. Does "Fig. 1" show this? This figure and its caption don't indicate as much.

Fig. 1. contained a sample image as filmed by each observatory, hence the reference. Both Fig. 1 and 2 have been altered (Fig. 2 contains now the sample images and Fig. 1 features more information on the observatory lay-out) and references to figures in the text we altered accordingly.

L. 223-225. and Fig. S1. Confusing. Consider removing at least the "days" from the text? As it reads now, the sentence references a Fig. with an x-axis in periods (which equal 6 hours), 18 hr periods, hours, days, and hours in multiples of 18. This is too much. Also, consider changing "*" to "x".

The days were removed and "*" has been changed to "x".

L. 247. I don't see how Fig. 2 demonstrates this point: it's a 2D schematic with no information about the substrate below the mobile fauna.

Figure reference has been removed.

L. 399. Add a sentence describing the diversities.

L.399: "This observation does not imply that the MAR is more diverse than the NEP since imagery only gives a partial overview of the actual diversity (Cuvelier et al., 2012)." When comparing samples, an overall higher diversity was observed in the Pacific when compared to Atlantic hydrothermal vent

ecosystems, with species richness being positively correlated with spreading rate, associated distance between vent fields and longevity of vents (Juniper and Tunnicliffe, 1997; Van Dover and Doerries, 2005). Nevertheless, such observations remain subject to how well a certain locality is studied and if all faunal size fractions (meiofauna to megafauna) are included in assessing diversity (e.g. Sarrazin et al., 2015). Diversity estimates represent one of the main limitations of imagery analysis which is limited to quantifying and correctly identifying (assessing) mega-and macrofauna (~mm).

L. 423. The assumption is the same individual is returning every time? Can you really say this?

Caution is needed to identify recurring animals as being the same individual between images. Though, here it appears to be the case. It is quite a recognisable animal (a large golden-coloured polynoid) which is not observed very often on imagery at the Eiffel Tower edifice. The size and number of scales seems to confirm that it is the same animal.

L. 425. And so?

The paragraphs for this section were restructured and succinct information on ecological interactions has been added. In this particular case, following sentence was added: "Many of the free-living polynoid species are known as active predators (Desbruyères et al., 2006) moving rather swiftly across the FOV looking for prey and were even observed attacking extended tubeworm plumes at NEP (Cuvelier et al., 2014)."

L. 435. What is the "very distinct spatial distribution in NEP"?

This was a reference to the heat maps published in Cuvelier et al 204. The reference has been added. The sentence reads as follows:

"Sea spiders showed a very distinct spatial distribution in NEP featuring a localised clustering behaviour (see heat maps published in Cuvelier et al., 2014), whilst their presence on the MAR was occasional."

L. 442-443. Unclear what the authors are saying here.

This sentence was rephrased to: "While being an abundant taxon with a localised clustering behaviour at the NEP site, it is scarce and vagrant at the MAR. Their niche occupation at the studied sites is likely to differ thus causing the discrepancies observed."

L. 487. How fast do mussels move? Did you expect to see a difference at a frequency of 6 hrs?

Species of *Bathymodiolus* have been observed moving 0.74cm per hour (Govenar et al., 2004). Hence, if they would start to move, we should be able to observe them moving away between 2 consecutive images or videos (6h apart) since the distance they could cover in 6 hours amounts to ~5cm and the distance from the mussel bed to the edges of the FOV equals 15-20 cm. Here we observed mostly mussel repositioning, no large distances (cm's) were covered.

L. 594. At vents or everywhere?

This sentence applied to our study, so we added "at hydrothermal vents".

L. 607. Review Lau back-arc basin hydrothermal vent studies linking faunal variations with environmental gradients.

Contrastingly to the works carried out in the Lau back-arc basin by Podowski et al. (2009) and Sen et al. (2013, 2014), where multiple measurements allow for extrapolations across a mapped surface and

more successfully link environmental gradients to faunal presences, our study relies on single point measurements. These single-point measurements make establishing direct links between faunal variation across the FOV and environmental variables hard, despite the high resolution of data available. Spatial variation in environmental gradients is high as illustrated by the Lau back-arc basin hydrothermal vent studies and even when examining relatively small surfaces as is the case here. For instance, at the Grotto edifice at NEP, next to the TEMPO-mini deployment, temperature arrays (strings of loggers) in two areas of ca. 30x50 cm on the faunal assemblages demonstrate the high spatial variation at cm-scale both in fauna and temperature (Lee et al., 2015). We realise that the statement in L607 might have come across as an over-generalisation. This was clarified in the text.

Discussion and Conclusion: Explicitly offer at least one mechanism to connect the influence of the tides and temperature, and the influence of tides and the pattern observed in tubeworm appearance.

The influence of tides on the temperature regimes has been discussed in L549-561. Temperature variability at hydrothermal vent on the Juan de Fuca Ridge was shown to correlate with the variability of the current speed and direction (more so than with ocean tidal pressure) (Tivey et al. 2002). Potential mechanisms causing tide-related variability in hydrothermal fluids include the modulation of seafloor and hydrostatic pressure fields by ocean tides, modulation of horizontal bottom currents by tides and solid earth tide deformations (Schultz and Elderfield, 1997; Davis and Becker, 1999).

A section was added on modulation of temperature by tides in section 4.3.

Extension retraction in tubeworms and possible links with environment was briefly touched upon in L495-496, but we added the following sentence "Emergence/retraction movements of siboglinid tubeworms were proposed to be a thermoregulatory behaviour or suggested to be governed by oxygen or sulphide requirements (Tunnicliffe et al., 1990, Chevaldonné et al., 1991) or tolerance to toxic components (sulphides, metals, etc.). Changing hydrothermal inputs (high sulphide concentrations/high temperature) and oxygen concentrations could thus regulate tubeworm appearances, reflecting the tidal patterns of these environmental variables."

Discussion and Conclusion: Do the authors believe the tides change the overall temperature of a vent, or just the outflow directionality of the fluid at the point location of the probe?

Based on personal observations on imagery, the fluid flow changes direction when currents are strong, no longer (temporary) bathing an assemblage in fluid flow. A negative (though not significant) correlation was observed between fluid flux and current speed at MAR (Sarrazin et al., 2014). In our opinion, the overall temperature of a vent does not change but the fluids get redirected following the currents and locally perceptions might change since a probe may be only periodically exposed to the expelled fluid.

Similarly, methane seepage was shown to be modulated by periods of enhanced bottom currents associated with diurnal shelf waves, internal semidiurnal tides, and also wind-generated near-inertial motions (Thomsen et al 2012).

Discussion and Conclusion: Were all the tubeworms alive? If not, what effect could this have had on the ecological patterns observed/not observed?

No, it is very likely that several tubeworms tubes were "empty" or no longer containing live individuals. Visible tubeworm densities ranged between 11-70% of the entire tubeworm bush at the

time points analysed. This will have no influence on the temporal patterns revealed, such as the tidal pattern. It could play a role in the spatial interpretation, e.g. dead tubeworm areas can be characterised by presence of certain organisms or a lack of associated organisms and thus be an indication of a changed microhabitat.

TECHNICAL COMMENTS

L. 56-60. Rewrite "Key questions" sections so that the sentences are grammatically correct. For example, ". . .put forward are: (i) are tidal. . ." and "the most? And finally, (iv) do . . .".

ok

At times, I found the writing was too informal for a scientific manuscript. I was not happy with the (repetitive) use of "vs.", "on one hand. . .on the other", and "and/or", and [L. 422] "...individuals appeared very attached..." The tense of the manuscript jumps around sometimes. For example, L. 134-135. There are many inconsistencies in the text formatting: *"Hours" was written as "hours", "hr", and "h", with a space or with no space between the number and the shorthand "hr" or "h". This inconsistency was even more confusing because the UTC time was also reported using "h" (again, with either a space or no space between the number and the "h") or UTC was reported with "AM" or with no units. *In-text citations are inconsistently formatted: "et al.," is often missing a comma; both "and" and "" are used for 2 author papers; multi-paper citations were not always listed chronologically [L. 406]; author's initials included [L. 164]; and missing a comma after authors [L. 186] *Values with units are reported with and without spaces. For example, m vs. m. *The shorthand for "Figure" is written with and without punctuation, within the text and the figure captions (i.e. "Fig." and "Fig"). *Section numbers are written with and without "." at the end (in the section titles, as well as when referred to in the text). *"Oxygen" or "oxygen". *Text jumps between "iron" and "Fe" in same paragraph. *Formatting the title of a subsection varied between: title in the text (e.g. L. 253 and title on a separate line (e.g. L. 341); indented or not; followed by a long/short/bolded/no dash. *Mean and stdev written: ± units, ± units, and ± units. *Within the same paragraph, reporting a date range changes from "date to date" and "date - date". *The Reference section requires some attention. For example, "Year" vs. "(Years)"; ending the authors list with a ","; inconsistent formatting of the volume number, issue number, and page text; inconsistent spacing; inconsistent punctuation; different color text [L. 839-840]?; etc.

The manuscript was checked thoroughly to remove such cases and other inconsistencies. Extra attention was given to the tenses used.

L. 165-166. Insert space

ok

L. 177. Remove "()".

ok

L. 198-200. Poorly written. Rewrite sentence.

Sentence was changed to: "No specific correlations between faunal densities and environmental variables were presented. The high spatial variation occurring at hydrothermal vents proved difficult to capture with the experimental settings from the 2011 deployments. The probes at NEP were placed at a distance from the filmed assemblage and the relatively large surface filmed at MAR decreased the representativeness of the single point measurements. The measurements made were thus more representative of an overall variability, not at the scale of individuals."

L. 237. Change "featuring" to "with".

ok

L. 272-273. (as one example) Watch the p-value sig. figs.; at times, they vary within the same sentence. Personal preference: never report p = 0 (or in this example, "0.00"), report it as p < 0.001.

P-values have been checked and changed accordingly.

L. 305. Reference Fig. 5 somewhere in the paragraph.

Ok

General: Write out values less than 10 (e.g. 9 months –> nine months)

Ok

L. 339. Repetitive.

This was omitted.

L. 344. Use "...was already.." or "...as well", but not both.

We deleted "as well".

L. 430. Correct. "...abundant on to areas..."

Corrected

L. 432. Correct. "...both species [are] considered..."

Corrected

L. 474. "...feeding [activity]..."?

Corrected

L. 494. Open bracket with no closing bracket.

Corrected

L. 496. Delete "Until now", because it still has not been established.

Corrected

L. 508. "...by a [longer] study..."

Corrected

L. 510. "...as they [become] more..."

Corrected

L. 522. "...in a single taxon..."

Corrected

L. 545. "...for both [temperature] probes..."

Added

L. 567. "...were close to..."

Corrected

L. 572. "...Tunnicliffe et al., 1997)."

Corrected

L. 614. What is meant by "harshness"?

The harshness of the local environment or rather the extreme environmental conditions and gradients. We changed the sentence to "Biotic interactions are at play as well; they can be observed thanks to the remote observatory set-up granting us access to long-term high resolution data (Matabos et al. 2015)." as it appeared more relevant to our study

L. 617. "...and [piloting] skills..."?

Added

L. 629. "This is [likely] due..."?

Accepted

L. 635. Capitalize "automated".

ok

Do they "need" to, or would it be helpful?

In any scenario, it would be helpful. However, if we want to increase the resolution and duration of analysis (and deployments), automated tools are needed because of the time-consuming character of imagery analysis. There is only so much a person can do in a certain amount of time. In order to reflect this issue, we changed the sentence to: "(Semi-) Automated tools should be developed for specific taxa and settings to assist in assessing faunal abundances on in images."

Suggestion: "faunal abundances [in] images."

ok

Figures (in general):

Consider

      (i) standardizing graph formatting throughout the manuscript,

      Fig. 4 was the only graph that stood out and was made consistent with the formatting of the other graphs

      (ii) removing repetitive information in graph titles (e.g., Fig. 6: use the probe name only vs. "T-MAR for imagery duration - hourly average"; that information is in the caption),

      (iii) clean up the axis ticks, labelling, and titles, and (iv) move footnotes (denoted by an asterisks, "*") at the end of a Fig. caption.

Fig.1. This figure is missing some key information. The text for the scale bars is too small. Why is there text and colour bars in the lower right-hand corner of the NEP bottom inset? Label Canada and/or

USA? Label the oceans? In the caption, explain or refer to the 4 insets. What are we looking at here? Consider providing larger photos? Add punctuation for "Fig. 1." and "Matabos et al.".

The sample images featuring the scale bars have been added to Fig. 2, thus increasing readability (see inserted figure below). Fig. 1 and legend have been changed accordingly including more information on the observatories.

[Figure]

Fig. 1. Location of the two study-sites in the Atlantic and the Pacific Ocean, along with some other well-known vent fields for reference purposes. The NEP inset (top) shows the location of the different instrumented nodes of Ocean Networks Canada at the right and the TEMPO-mini ecological module deployed at Main Endeavour Field on the Juan de Fuca Ridge (NEP). With the MAR inset (bottom) representing the Atlantic observatory (MoMAR) at Lucky Strike vent field on the left and the TEMPO ecological module on the right. For more details of the exact location of the observatories within the hydrothermal vent fields see Matabos et al. (2015) for MAR and Cuvelier et al. (2014) for NEP.

Fig. 2. Is it necessary to retain some transparency (the key colours really do not match the colours overlaid as semi-transparent)? Change to "Microbial [c]over" (in key). Why is the text "Ifremer" in the bottom right corner and why is it coloured in as "Pycnogonida" (in yellow)? The hatching in the MAR image (for "Mussel background" is difficult to resolve. Add punctuation for "Fig. 2.". Move footnote to the end of the caption?

The semi-transparent colours allow the reader to see what is underneath and could facilitate interpretation.

The Ifremer text is part of the watermark on the images recorded, as is Neptune Canada written before it. Since an original sample image has been added to this figure, we hope it is clear that it indeed is part of a watermark and not a pycnogonid patch.

[Figure]

Fig. 2. A sample image as recorded by the ecological observatory modules for MAR and NEP (top) and a map of the fields of view (FOV) featuring the various taxa assessed (bottom). Taxa or other features that are shared between the two have the same colour codes. Gastropoda applies to Buccinidae for NEP and bucciniform Turridae on MAR. White substratum could possibly be anhydrite with encrusted microbial mats. 'Mussel background', 'background' and 'probe' correspond to areas that were not assessed because of increased distance to the camera's focal point and associated light emission and were therefore not included in the surface calculations. The white arrow represents the fluid flow exit and direction. No visible emission was observed on NEP. Visiting fish and crab species were not included (Table 2). Crab presence on MAR tends to correspond predominantly to shrimp distribution (Matabos et al., 2015). Surfaces filmed and analysed are listed in Table 2. '*' is a shared taxon but not visible on MAR due to the scarce presence and low densities.

Fig. 3. Graphs and text are grainy. Consider deleting "densities" from the 10 individual titles (repetitive), and just list the taxa. Reduce the number of x-axis ticks (I can't tell which line is associated to the values listed). Add y-axis title. The Crab graph is missing the number "10" on the y-axis. Shorten the number format of the y-axis for the MAR Pyncognoid and Shrimp graphs (i.e. 0, 1, 2... vs. 0.0, 1.0, 2.0...). Mention "23 days" in the caption. Change to "...with an "*"" OR "*Taxa with significant trends."

Resolution of the original graphs which will be submitted separately is better than those in the pdf of the manuscript. Axes, labels and (sub)titles have been changed. Instead of reducing the number of ticks, we chose to elongate those ticks that correspond to the numbers below.

Fig. 4. When printing in black and white, it is impossible to tell the difference between light blue and light gray. In Fig. 5, NEP is black and MAR is light gray (which can be distinguished in black and white print). To standardize the figures, and for printing purposes, consider changing NEP to black and MAR to gray for Fig. 4. Consider rewriting caption and/or changing the x-axis title. I'm not sure what the value is supposed to be, hours or periods? Reorganize so the sentence doesn't start with "

We agree and changed the colours of the graph to correspond to the other graphs.

Fig. 5. To save space, consider adding a 2nd axis to the temperature graph (to display both NEP probe temperatures, instead of repeating the MAR data. Remove "shortterm" for graph titles? If not, change

to "NTU short-[t]erm". Is it necessary to repeat the same key for 3 of the 4 graphs? Although this is not the only time the figures include stacked graphs, this is the only time the x-axis is included.

Changes have been carried out.

Fig. 6. Shorten the y-axis labels to represent a count of the days (e.g., day "1", "2"..."23" vs. "2011-10-07", "2011-10-08"...). Add titles for the x- and y-axis (e.g., "day" and "hour"). Indicate somewhere in the figure or caption: temperature in ∘C.

The comment on changing the dates to count of days is opposite to what was proposed for Fig. S3. We opted to stick with the dates because they allow a reference to the time series investigated

Other changes have been carried out.

Fig. 7. More information is required for the caption. Are there gray and black vertical lines (appear to be)? If so, what do they represent? What are the 2 blue dashed horizontal lines on each graph? Change "X-axis" and "Y-axis" to "x-axis" and y-axis (to be consistent with text).

Vertical lines are all in the same colour. The horizontal blue dashed lines indicate the point of statistical significance (here ACF=0.8, with p<0.05), with the lines above towards 1 and below towards minus 1 being significant (this was added to the figure legend).

Fig. 8. Label and mention: one graph is MAR and the other is NEP. Change text and vector lines to black (vs. blue). Difficult to read the text on the graph, increase the size? Define RDA? There is a noticeable difference in the size and quality of text in the left and right graphs. Standardize?

Changes have been carried out.

Fig. 9. Is the x-axis in hours? Include "Temperature (∘C)", not just "∘C" for y-axis. Remove redundancy in the graph titles and consider adding this information to the figure captions, e.g. "...over six and nine months".

Changes have been carried out.

Fig. 10. Confused again by the x-axis title and the caption. This data is for a one-week period equalling 200 hours, but the x-axis title is "Period", not "Hours", and plus, 1 week = 168 hours. Please clarify. Change the lines to black (no need to be coloured red).

Lines are rendered in black.

Fig. S1. Include the information for the white vs. black symbols. Why change the x-axis intervals? If each period = 6 hrs, and there are 45 periods, the graph represents 270 hrs or 11.25 days. Include this easy to understand temporal reference (and why this length of time)? Change the lines to black (no need to be coloured red).

Information on the black squares was added (Black squares indicate periods significant at the 5% level.). X-axis interval changed because the time series for fauna is recorded at a 6h frequency while the temperature time-series as presented in Fig. 10 was recorded on an hourly frequency. This is mentioned in the figure legend as (1 period on x-axis=6h).

The periodograms have a maximum length of n/2 with n being the length of the time series analysed. For fauna this is thus 23 days/2. Length of the time series was added to the legend.

Lines are rendered in black.

Fig. S2. Similar concerns to Fig. S1. Why change the x-axis intervals? Remove the repetition of the x-axis title (i.e. only include "Period" once). Change the lines to black (no need to be coloured red).

See reply above. X-axis label was only included once.

Fig. S3. Change to "...(a) MAR and (b) NEP...". Are the "random" data consecutive? Yes

Why not report the specific month and year (even if it was selected randomly)? In analogy to what was decided for Fig. 6. We chose to add the specific dates to the random time series.

Use the same style quotation marks at start and end of the quote -or in this case, consider not using quotation marks at all. Many of the same comments and concerns as expressed for Fig. 6.

Tables (in general):

Consider

      (i) reducing the number of lines (vertical and horizontal) for each table,

      (ii) removing repetitive information (e.g. "2011-2012"),

      (iii) condensing the area of each table (there is often a lot of blank space between rows),

      (iv) use either "Table :" or "Table .", but be consistent, and

      (iv) move footnotes (denoted by an asterisks, "*") below a Table.

Table 1. "[o]xygen". "[T]wice". Use "NA" instead of "/" (or define "/"). Be consistent, "min" or "min.". If minutes = "min", seconds could = "sec". As in the text, include "at" when listing the sample times (e.g., at 2h, 6h...UTC). Be consistent with apostrophe symbols for the coordinates (styles change between MAR and NEP). Explain/include row title for "Wireless" and "Cabled".

Changes have been carried out.

Table 2. Move footnote to below table (or at least the end of the caption). "[A]re visiting...". Fix: "see fig. X"? Reverse how the gap range is reported ("9 to 93 gaps" and "5 to 93 gaps")?For surface filmed and surface analyzed, be consistent with sig. figs. and with the information provided (why list the ca. dimensions 2 out of 4 times?). Reported frequency as "6hr" and "12 h" in the same table (use a consistent format). Check citation formatting (missing punctuation). Use "NA" instead of "/" (or define "/"). The lines of this table are bolded, why?

Gaps have been defined in the text and the legend (see comment above) and equal 9 for MAR and 5 for NEP. 93 is the amount of images theoretically present in our 23-day time series at a 6h frequency. They have been included as follows: "93 total with 9 gaps". Other changes have been carried out.

Table 3. n = ? (photos?) Missing "h" after "553" twice. Add a space to "( 2 days)".

n=number of images, other changes have been carried out.

Table 4. Include "∘C" in the table caption and remove it from each record. "[S]tdev"?

Changes have been carried out.

---

## Author Comment (AC3) · 15 Feb 2017

**Referee 1**

We would like to thank the reviewer for a thorough revision of the manuscript. Several of the issues raised improved the coherence of the manuscript. We addressed all points raised by the reviewer below.

General comments: The paper by Cuvelier et al. is an interesting study that uses time series analyses, conducted concurrently at two different hydrothermal vent settings in two different oceans. It is a unique study that deserves attention and it is good to see such work being done. However, there are some important scientific issues that need to be addressed. A major finding of the paper is that patterns in temperature and tubeworm behavior were seen at both the Pacific (NEP) and Atlantic (MAR) sites that correspond to 6 hour time intervals, which the authors conclude is linked to tidal patterns. Additionally, they note that the same effect is seen 6 hours apart between the two sites which is a product of the time difference between the two sites.

The 6 hour periodicity might be present, however, the link to tidal patterns is not sufficiently developed. There is no data on the tidal rhythms or whether the increases or decreases in tubeworm appearances or temperature values correspond to specific events of the local tidal patterns. In order to come to the conclusion that the periodicity seen in this study is indeed linked to the tides, tidal data needs to be examined and presented within the context of the results of this study.

Data on local tidal patterns and how they correspond with the rhythms found in temperature was indirectly included by means of pressure in supplementary figure S3. Potential mechanisms causing tide-related variability in hydrothermal fluids include the modulation of seafloor and hydrostatic pressure fields by ocean tides, modulation of horizontal bottom currents by tides and solid earth tide deformations (Schultz and Elderfield, 1997; Davis and Becker, 1999). Though the modulation of temperature by tides at several hydrothermal vents on the Juan de Fuca Ridge is thought to be mostly indirect through bottom currents (Tivey et al., 2002). A paragraph on what is known for local tidal patterns at our 2 sites of interest in section 4.3 is added. Multiple day periodicities present in the temperature data have already been linked to local oceanographic patterns from the literature in the discussion e.g. L559-561

The paragraph incorporated is as follows:

"Tidal rhythms observed in the temperature time series for NEP and MAR are concordant with observed tidal signals for the region. For instance, in the North-East Pacific, measured tides in the Barkley Canyon, another instrumented node from ONC closer to shore, were mixed semidiurnal/diurnal at 870m depth (Juniper et al. 2013). In the same canyon, periods of enhanced bottom currents associated with diurnal shelf waves, internal semidiurnal tides, and also wind-generated near-inertial motions were shown to modulate methane seepage (Thomsen et al 2012). While, temperature variability at hydrothermal vents at Cleft Segment on the Juan de Fuca Ridge was shown to greatly diminish when current directions did not shift in direction with the tides, it was suggested that the modulation of temperature by tides was only indirect, through the modulation of horizontal bottom currents (Tivey et al., 2002). These horizontal bottom currents showed 12.4h tidal periodicity which was also found in the temperature time series of the aforementioned article as well as in our NEP temperature time series. Patterns in temperature variation of the MAR time series correspond to the tidal signal observed in the Lucky Strike vent field at 25h and to the semi-diurnal tidal oscillation at 12.30h (Khripounoff et al., 2000, 2008)."

A sentence on three mechanisms explaining tide-related variability (i.e. the modulation of seafloor and hydrostatic pressure fields by ocean tides, modulation of horizontal bottom currents by tides and solid Earth tide deformations (Schultz and Elderfield, 1997; Davis and Becker, 1999)) was added as well. It appears from the results, that by and large, not a lot of changes overall were seen. The mussel and shrimp densities at MAR and the pycnogonid densities are the only ones that show an increase over time. This brings up a number of issues and considerations that ought to be treated in the discussion of the paper. For example, one major issue is the spatial extent: the areas analyzed are very small and the authors should include a discussion of the spatial scales at which appreciable changes in the megafaunal community can be observed. In the cases of the increases in densities of taxa, it is surprising that the discussion includes no references to successional patterns. The authors do mention that the mussels represent a climax community at shallow Atlantic vent sites, but there is no discussion of recruitment or colonization as being possible explanations for the observed increases in densities. And, the overall stability is not discussed very well either. Though there is a brief reference to differences in the level of dynamism in vent communities being possibly linked to spreading rates, this is not discussed very much despite stability being one of the major findings.

With this manuscript we intended to focus on the comparison between the two sites rather than on the high local spatial variation observed at each hydrothermal vent site. Using the same reasoning, we limited ourselves in the large ecological implications and extrapolations for successional patterns since we are aware that the FOV is rather small and shows only a single assemblage, while hydrothermal edifices are inhabited by mosaics of different faunal assemblages. These issues were more thoroughly explored in Cuvelier et al 2014 and Sarrazin et al 2014 and therefore were not mentioned as such in the current manuscript, though references to these papers were used. Our long-term experience with imagery data (from Sarrazin et al. 1997 up to now) have shown that the spatial scale of observation we are using in our observatories is sufficient to observe changes in composition and abundance of visible taxa. For example, a study by Cuvelier et al. (2011) showed that on the Atlantic Eiffel Tower edifice, the overall percentage of biological colonization and mussel coverage were stable on a decadal scale but that on shorter time scales as well as on smaller spatial scales, significant differences in microbial cover and individual assemblage coverage and distribution were observed.

However, taking into account that the current manuscript should be able to stand alone as an independent study as well, we addressed the issues raised by the reviewer by adding a section on the spatial variation issues and hence limitation of extrapolations at the beginning 4.1. of the discussion linked with observed stability.

Section added: "The two observatories filmed one single assemblage over time, whereas hydrothermal edifices are characteristically inhabited by mosaics of different faunal assemblages, spatially distributed according to local environmental conditions and microhabitats (e.g. Sarrazin et al, 1997; Cuvelier et al 2009; 2011 Mar Ecol, Sarrazin et al. 2015). High local variability in environmental variables on a scale of centimetres and steep physico-chemical gradient contributes to these patterns (Sarrazin et al., 1999, Le Bris et al., 2006). Differences in spreading rates may influence community dynamics at vents by creating less habitat stability in higher spreading rate settings (Tunnicliffe and Juniper 1991, Shank et al. 1998). While relative stability in faunal composition has been observed on a number of edifices, even reaching decadal-scale stability at some (e.g. Eiffel Tower), smaller scale variations, both in space and time, do occur (Cuvelier et al 2011 L&O). Hence, the variations in faunal densities observed during this study may not apply to the hydrothermal edifice as a whole; the presence of tidal rhythms in the organisms and in temperature, even though observed on a smaller surface, are likely to apply for the entire hydrothermal structure."

Regarding the lack of discussion on colonisation and recruitment, 23 days in one single year is a rather short window in time to be able to observe colonisation and recruitment, even when continuous recruitment is assumed. Also, it is important to bear in mind that new recruits are small and inconspicuous and can go easily unnoticed, especially when using image analysis. Recently, succession has been observed at the NEP observatory over a period of ~one year, with the formation of a small flange, colonised by *Paralvinella sulfincola* and followed by the rest of the community (unpublished data, see Sarrazin et al. 1997 for succession patterns). A short note on the fact that 23 days appears too short to allow observation of succession patterns was added to the manuscript.

The writing itself needs considerable improvement. First, it should be read by a native English speaker since there are a number of grammatical errors and sentences that appear to be lost in translation.

We have thoroughly checked the manuscript for errors also taking into account the comments of the second reviewer, whom specifically pointed out the sentences that were poorly written. These were changed accordingly. We think that this approach considerably improved he manuscript.

Secondly, the discussion, particularly the part with reference to the different taxa is written as a list of short, highly abbreviated paragraphs. This needs to be improved upon, restructured and rewritten so that a cohesive story is presented as opposed to a list of short comments. For example, paragraphs should not end with a new thought or idea such as line 432, on page 12 which states 'Both species were considered predators or scavengers.' This is an important aspect to the biology of the snails discussed within this paragraph, without a doubt, but it is something that should be expanded upon, and should not be the final, concluding sentence of a paragraph that up to that point has not made any mention of trophic relationships or feeding biology. As it stands now, this part of the discussion reads basically like bullet points instead of a cohesive discussion.

This part was kept succinct on purpose, in order to avoid an extensive discussion which would appear more like a review than a research paper. Since this was an issue both reviewers touched upon, sections 4.1.2 and 4.1.3. were slightly restructured, though main lay-out was kept, and paragraphs were elaborated into a more cohesive text, mentioning relevant ecological interactions.

E.g. from section 4.1.2: "Many of the free-living polynoid species are known as active predators (Desbruyères et al., 2006) moving rather swiftly across the FOV looking for prey and were even observed attacking extended tubeworm plumes at NEP (Cuvelier et al., 2014). Free-living MAR scale worms were preponderantly associated with bare substratum, while those quantified for NEP were only those observed on top or within the tubeworm bush. They were also visible on the bare substratum surrounding the tubeworm bush but this area was not taken into account during this study. While there was a difference in substratum association between polynoids as observed by the two observatories, all individuals seemed to be rather territorial (see Cuvelier at al., 2014). On the MAR, one individual appeared to repeatedly return to one single area within the FOV after excursions. Such behaviour might be indicative of topographic memory and homing behaviour. "

Specific comments:

**Introduction:**

In the key questions in the last paragraph: the first question is 'are tidal rhythms discernible in both vent settings?' It would be better to perhaps say 'are rhythms discernible in both vent settings that correspond to tidal patterns?' Since making the actual connection between the patterns seen in this study and tides is beyond the scope of the study.

**We changed the first que question to: "Are rhythms discernible in both hydrothermal settings?" since we searched for rhythms and one of the main results was the correspondence to the tides.**

The introduction should include some background about the major faunal groups and community structure at the two study sites. This is presented currently in the Methods section and certainly more details can be presented there, but the Introduction should also contain this information because understanding the settings is important contextual information.

We added that the MAR is *Bathymodiolus* mussel dominated site while NEP is a *Ridgeia* tubeworm dominated site. However, since the current paper already counts 17 pages of text (figures, tables and references not included) we do not want to repeat the same information both in introduction and methodology.

Methods and Results: A number of key methodological information is missing. Though it is mentioned that the MAR observatory was positioned to face the Eiffel Tower edi- fice, no such information is given about the NEP observatory, such as whether it is also facing a chimney structure or not. If it is also placed facing a chimney structure, then this should be clearly stated early on in the manuscript, because chimney communities differ from areas of diffuse flow (and even host different morphotypes of Ridgeia tubeworms) which would mean that this study is examining chimneys on vents from two different oceans, which is very specific.

This information is present in L76-77 for NEP.

We added a little bit more information for both deployment sites:

MAR: "TEMPO was positioned at 1694 m depth at the southern base of the hydrothermally active Eiffel Tower edifice, a large 11m high edifice."

**NEP: "It (TEMPO-mini) was deployed at a depth of 2168m on a small 5m high platform on the north slope of the Grotto hydrothermal vent, a 10m high edifice at Main Endeavour Field (MEF)."**

It is not mentioned, but clear from the photos, that the camera is positioned facing forward. In this case, there has to be clear details on how the spatial extent of the field of view was calculated. This is very important information and I am surprised that it has been left out. Other details about the imagery is also missing, for example, since video cameras were used, I assume that video stills were taken at the appropriate time points and those video stills were analyzed and used for marking the animals (in which software?), but these details are not present in the manuscript.

A section clarifying these methodological details has been added to 2.3 Short temporal analysis: "For the 23-day period, a screen still was taken every 6 hours at 0h, 6h, 12h, 18h UTC. For each site, these screen-stills were used as a template in Photoshop© to map and count faunal abundances. Surfaces were estimated by using known sizes of sampling equipment during deployment. Faunal densities were quantified at a 6h frequency, while for one image every 12h the microbial coverage was assessed, both for the analysed area (see Table 2). To pursue the latter, the microbial cover was marked in white and the rest of the image rendered in black. Using the "magic wand tool" of the ImageJ image analysis software (Rasband 2012), the surface covered by microbes was quantified and converted to percentages."

I think that it is inappropriate to use tubeworm abundances or tubeworm densities since in reality, what was counted where the extended plumes. Throughout the text, this should be changed to visible plumes or extended plumes, etc. and not tubeworm density.

This was specified in L149-150 of the methodology section. This issue will be addressed by adding a sentence stating that "from here on tubeworms visibly outside of their tubes will be referred to as tubeworm densities".

In general, density should not be used at all. In both cases, the surface filmed and analyzed is considerably less than 1 m2 which means that all the density numbers are extrapolations and I don't think that is appropriate. Since within a site, the same area is filmed and examined for all 23 days and time points, the use of numbers of individuals instead of extrapolated densities would be more appropriate.

We chose to work with densities in order to use relative values as a standardisation. Moreover, within the NEP time series, the zoom changed twice. Even though it was a minor change, the surface of the FOV changed and thus densities were preferred in order to allow comparisons amongst the images of the NEP alone (see Cuvelier et al 2014 PlosOne). When comparing MAR and NEP, the difference in surface filmed and analysed (see Table 2) was quite large, hence densities were used to mitigate the sample size which in this case is the FOV. This is also why we used the percentage coverage instead of area covered in square cm or m. Finally, the use of densities and % are the only way to allow comparison with other data series.

Similarly, for microbial mats, use area coverage instead of percentage of area (and was percentage and density calculated based on filmed area or analyzed area?)

All densities and coverages were calculated for the analysed area (hence the name choice, this is explained more clearly in the added methodology paragraph mentioned above), other areas were not taken into account as stated in L128-130, this also applies to the microbial mats.

There is no explanation as to why areas of microbial mats were examined at 12 hour intervals and not at 6 hour intervals like the fauna.

Variation observed in microbial mat coverage was rather low at a 12h frequency, which was why we decided to not increase its resolution. The chosen resolution is sufficient to observe coverage changes.

Due to the difference in depths and ambient temperatures between the two study sites, raw temperatures should not be used at all. Instead, rescaled temperatures (raw temperature – ambient temperature) should be used and presented. The authors even say that there is a 2 degree difference in ambient temperatures between the sites and they say that even when this is taken into account, the NEP temperature recordings have a higher mean and maximum temperature. However, that does not mean that the distributions are necessarily different. A simple t test should be done to test if they are significantly different or not. The temperature data shown, for example, in Figure 5 seems to indicate that they are not significantly different since they appear to basically differ by about 2 degrees, which is the difference in ambient temperature between the two sites.

The differences between raw and rescaled values are presented in Table 4 + L280-285 and discussed in L535-539.

We opted to use the raw values for representation and analyses purposes because it represents the temperature the animals experience at the MAR and NEP. In addition, using raw or rescaled temperatures has no impact on the identification of rhythms or lags between the two sites. In case

one would calculate the amount of hydrothermal fluids based on the temperature, it is better to use rescaled temperature for comparison.

To resume, raw temperature = temperature experienced by the organisms, rescaled temperature = proxy of the hydrothermal input.

T-tests compares the means of the two time series analysed. T-test were significant for all combinations (T602-T603, T602-MAR and T603-MAR) at p<0.05 for the rescaled values. However, when looking at the boxplots the differences in variance in the time series are significantly less distinct. Since we were more interested in the variations over time, instead of the mere differences between NEP and MAR, an ANOVA seemed more appropriate. This test analysed if the variance occurring within a time series is larger (not significant) or smaller (significant) than that observed between the two time series. Anova's revealed no significant differences between MAR and NEP (T602 and T603) (p>0.05). These results will be incorporated in Table 4.

The other major issue I have with the manuscript in its current form is the use of statistical tests. Some of them are not quite appropriate and others can be tweaked.

I am not convinced it is appropriate to use a linear regression model to state if changes in densities over the 23 day period were significant or not. The independent variable is time, which is actually specific time points. It is important to have Figure 3 to show the trends, but fitting a line to these data and using that to say the changes are significant or not is, I believe, incorrect. The buccinid density graph really illustrates this, where the densities increased, then decreased and then increased again. That clearly does not mean that overall, in the study time period, buccinid densities showed a decrease, or should be represented by a downward sloping best fit line (as it is in the paper).

We accept these comments for the bucciniforms and pycnogonids for the MAR, since their abundances reflect presence or absence. The trend lines for these 2 taxa were removed, however, the use of regression on the other data is appropriate. These regressions are used to describe the trends observed for the timespan observed (23 days), not to extrapolate to larger time scales. Hence trends were withheld for the remainder of the taxa.

The differences in analyzed areas between the two study sites needs to be considered very carefully. I understand that the setup could not accomplish getting the same spatial extent for the fields of view, certainly, that would have been near impossible to achieve. However, when comparisons are made, for example, in the discussion about pycnogonid densities differing greatly between the two study sites, this difference in FOV extents needs to be kept in mind. In fact, it would be very difficult to constrain whether differences in densities or numbers of a specific taxon between the two study sites is a real difference or due to sampling artifacts. Therefore, such discussions need to be treated very cautiously.

Indeed, it is impossible to accomplish similar surfaces covered by both modules. The TEMPO and TEMPO-mini modules are deployed before they are connected and activated. Once in place and connections (wireless or cabled) established, images are checked and small changes can be done by zooming in or by nudging the module or the camera with an ROV arm to slightly alter the FOV.

That is why, in order to compare differences between the 2 sites, without bias of the surface analysed and filmed, we used densities. Based on the knowledge existing for the taxa present at the two vent fields as discussed in section 4.1., we tried to describe the bigger picture by elaborating on the role of the taxa within the edifice community. For example, the pycnogonids at the 2 study sites

do present a different behaviour, i.e. clustering at NEP vs. single individuals visible at the edge of the mussel assemblage. Snails are also far more abundant on the NEP than on the MAR. We conveyed these differences between similar taxa in the different oceans more clearly in section 4.1.

There are some inconsistencies in terms of what was analyzed. For example, anemones are mentioned in the text, but are not in Table 2 which lists all the animals analyzed. Similarly, in the results (lines 229), mention is made of ophiuroids, which are not mentioned anywhere else before. And line 232 talks about a fish, which is also mentioned in Table 2, but was actually not seen in the stills, but in other video footage, which means, it was seen at other time points. Discussion of trends seen outside the time points relevant to this study should be discussed separately because it is has the potential to introduce bias (large, flashy fauna are easily seen and focused on). Limpets are mentioned and it is also said that they were not quantified (understandably so, because they are very small and numerous), but they are not shown in Figure 2.

Anemone densities did not change over time (L145-146 + L234-235) and were thus not assessed on a 6h frequency which is why they were initially left out of the Table. However, we followed the reviewer's advice and added them in Table 2 (see below).. We agree on the fact that the ophiuroids should be mentioned earlier on in the manuscript, e.g. in section 231, following L146 and they have been added to the table as well.

Observation of *Cataetyx* fish were more easily visible on the video sequences (moving imagery). On stills, they were difficult to observe due to shading and position within the FOV (more towards the back – in the background area of Fig. 2): If we would adjust brightness and contrast and different colour levels etc., the fish would be visible on the 6h frequency screen stills as well. Their behaviour is discussed separately as a visiting species.

Appearance of other fauna on other time points are limited to Zoarcidae and Majidae at the NEP (L478-485). These were not present on the 6h frequency screen stills and were thus not included in the analyses and figures and do not introduce bias. The mention of these taxa was linked with another comment addressed by the reviewer and we introduced them all in the methodology section, hence this was altered accordingly in the revised manuscript (in text and table 2 – the latter is included below).

Organisms shown in Figure 2 were those quantified alongside features necessary for interpretation (as mentioned in the legend). Limpets were thus not shown in figure 2. They are present as several strands that are fairly easily to locate, but there are also quite some individuals scattered across the tubeworm bush which makes it nearly impossible to add them to Fig. 2.

In general, the results and the discussion appear to have three major themes that should be dealt with in separate sections. The first is spatial trends and associations between taxa within each study site, the second is comparisons between the two sites and the third is temporal trends. These are often intermixed and the paper would benefit by having them discussed separately. There will be some overlap between them, but currently, the results and discussion comes off as being very patchy and leaping from one point to another, without complete development of each point. Splitting into different sections might help to make the paper more cohesive.

In our opinion, the discussion features two themes: comparison between sites at any given time (1) and over time (2). The spatial part is less important. We propose changing the subtitles in the discussion to make it more coherent to the results, as follows:

**Discussion**

4.1. Faunal assemblages => Comparison in faunal composition

4.2. Behavioural rhythms and variations => Short term variations and rhythms in fauna and environment

4.3. Environmental rhythms and conditions => Long term environmental variations and rhythms4.4. Limitations

The discussion was modified accordingly.

I suggest adding two figures or analyses: first, in addition to figure 3, which shows densities plotted for the different time points, the authors could benefit by having a similar figure, but with difference in numbers from the previous time point (6 hours) on the x axis instead of numbers.

Figure 3 has the 6h frequency on the x-axis, we realised that the label of the X-axis was not shown in the figure but this has been added. We also changed the legend of the figure as to convey this point more clearly. In our opinion, the addition of the figure suggested by the reviewer will show the same data be it in a different way and would thus be redundant.

Secondly, I strongly suggest having a figure with tubeworm appearances (and anything else that shows the 6 hour pattern) vs. temperature. And in fact, regression models could be applied to these and it would strengthen your case that temperature can be used to predict tubeworm behavior.

While an interesting figure and approach, the suggestion made by the reviewer would have fit better in the already published paper on the hourly NEP analyses (Cuvelier et al 2014). This type of figure has no temporal component which is the main scope of our study and was, since the current manuscript already features 9 figure and 3 supplementary figures, left out for now.

The discussion about the same taxon inhabiting bare substrate at one site but not at the other is very problematic, because the FOV for NEP does not include bare substrate at all. In fact, the caption for Figure 2 even lists bare substrate as being an MAR only feature. If bare substrate is not present in the images of NEP, then it is not possible to say that NEP taxa that are seen on bare substrate at MAR are not seen on bare substrate at NEP.

Bare substratum is visible surrounding the tubeworm bush at NEP, however, quantifying the animals present on this type of substratum proved impossible due to its increased distance from the camera. It was called background in figure 2 because there were also patches of microbial cover and individual *Ridgeia* tubeworms visible, hence not classifying as bare substratum *per se*. Some general observations could be made see example of polynoids in L418-420. A more adequate definition of the term background ("background were areas that were not assessed because of increased distance to the camera's focal point and associated light emission and were therefore not included in the surface calculations"), clarifying this issue, is included in the methodology section and the legend of Fig. 2.

In the discussion, certain taxa names are introduced for the first time, e.g., Bythograei dae, Bythitidae, and Majidae. These names do not appear in the Introduction or Methods, even when the animals are being introduced and they do not appear in Table 2 which lists the animals studied. The manuscript would benefit by keeping reference names for taxa consistent throughout the manuscript. See comment above addressing these issues: we accept the need for consistency in naming taxa throughout the manuscript and introducing them all in the methodology section, hence this is altered accordingly in the revised manuscript (in text and table 2 – the latter is included below).

The first part of section 4.2, ie, the discussion about mussel valve openings is problematic. By opening valves, do the authors mean that one of the siphons are visible and extended or simply open? Mussels filter water through their inherent and exhalant siphons and fully opened valves are generally only seen in sick or dead individuals. Therefore simply talking about mussels valve openings does not seem appropriate, or should be explained further.

**Yes, mussels with open valves are those with siphons showing, this change has been made.**

I do not know what software was used to mark and count the animals, but if the animals were physically marked, then it might be a good idea to examine the extended tubeworms more closely to see if there is periodicity in appearances among individuals.

See comment above addressing the lack of methodological details of the image analysis and the paragraph added to section 2.3. Individuals were marked manually in Photoshop on screen still templates.

For examples, are half the worms extending out of their plumes at a certain time while the other half remain in their tubes and at the next interval, do you see the retracted ones extended and the extended ones retracted, or is it random in who is retracted or extended at any time point?

Visible *R. piscesae* densities, i.e. individuals outside their tubes, based on an hourly analysis frequency ranged from 1038 to 8980 ind/m2, adding up to 8.5–78.6% of the total tubes that constituted the filmed tubeworm bush (Cuvelier et al., 2014). Applying these numbers to the 6h frequency analysed here, this adds up to 11-70% of the entire tubeworm bush. However, these numbers do not take into account possibly dead tubeworms or empty tubes.

We agree that it would be very interesting to monitor individual tubeworms and their extension/retraction rhythms. Individual annotation and follow/up needs to be done manually, since their relative positioning in the FOV changes due to small changes in the zoom during the time period, which represents a huge time-consuming effort.

A separate project was initiated to try and automatize the analysis of extension/retraction behaviour of individual tubeworms. This resulted in a preliminary paper (Aron et al 2013), that aimed at identifying the tubeworm openings, which would then, in a subsequent step, be marked as open (retracted tubeworm) and closed (extended tubeworm). Only a limited number of individuals could be detected automatically.

Aron M., Cuvelier D., Aguzzi J., Costa C., Doya C., Sarrazin J, Sarradin P-M (2013). Preliminary results on automated video-imaging for the study of behavioural rhythms of tubeworms from the tempomini ecological module (Neptune, Canada). Instrumentation Viewpoint, (15), 35-37.

When talking about periodicity of the more mobile animals like pyconoginids and snails, etc., it is important to keep in mind the time and spatial scales: Currently, I don't think it has been shown conclusively that the observed periodicity is real periodicity and not the result of mobile animals moving in and out of a small area of focus at their own individual paces.

We agree that it is not conclusively shown, since there are no significant periodicities or links with environmental variables, only indications as such. Their link with tidal periodicities would indeed depend more on their mobility, in the sense that they can move into an area when local conditions are favourable, e.g. when a region is temporary (not) exposed to fluid flow due to tidal currents.

The presence of pycnogonids over time within the NEP FOV is fairly restricted to a particular region (see Cuvelier et al., 2014 PlosOne). Few organisms were seen wandering around beyond this patch. Individuals that move around at larger distances are a large minority vs. those present at the specific spot (few individuals vs. >20), hence the periodicity observed is influenced by the less mobile "resident" individuals.

For the snails, however, no distinct region was occupied even though the species occurring a NEP tends to show more of a clustering behaviour (Martell et al, 2015) while those at MAR are more single occurrences. Buccinids are far more abundant at NEP than the bucciniform Turridae at MAR, making it nearly impossible to deduce any periodicities for the latter.

Though there is information on and a discussion of the CHEMINI system for measuring iron, no discussion or mention is made of sulfide. This is a very big gap in the discussion since sulfide is the fuel for the chemosynthesis based animals, and also a determinant of other animal distributions due to its toxicity. I understand that there was so sulfide sensor and therefore real sulfide measurements were not possible. However, temperature, oxygen and iron are correlated with sulfide and can be used as a proxy to a certain extent for sulfide. Even if real concentrations of sulfide are not included, sulfide itself should be discussed because it is the source of energy in this system and one of the main reasons why tubeworms extend out of their tubes.

As mentioned by this reviewer, the CHEMINI system can also be used to detect total dissolved sulphide. However, the CHEMINI used for the determination of total dissolved sulphide was not chosen for a long term deployment because the standards are not stable for a long period.

The current manuscript is very results oriented and therefore discussion of sulphide was not included, since it was not measured. With regard to its influence on the fauna, we chose to refer to the generic use of fluid or nutrients as such throughout the manuscript (and more specifically in 4.2.) instead of specifying sulphide concentrations or any associated metals. Temperature, Fe and Sulphide can be used as proxies for one another since they are positively correlated. This is now mentioned in the discussion section.

Temperature is a proxy of hydrothermal fluid input, not only of sulphide and Fe concentrations. However, it is yet impossible to decorrelate the role of each chemical compound, hence we can only discuss the input of O2 with cold seawater vs. the inputs linked to the fluids, i.e. those involving the provision of reduced compounds necessary for chemosynthesis and of potentially toxic compounds such as metals, sulphides and radionuclides.

Regarding the influence of sulphide on tubeworm behaviour, the following sentences were added in section 4.2. "Emergence/retraction movements of siboglinid tubeworms were proposed to be a thermoregulatory behaviour or suggested to be governed by oxygen or sulphide requirements (Tunnicliffe et al., 1990, Chevaldonné et al., 1991) or tolerance to toxic compounds (sulphides, metals, etc.). Changing hydrothermal inputs (high sulphide concentrations/high temperature) and oxygen concentrations could thus regulate tubeworm appearances, reflecting the tidal patterns of these environmental variables."

**Technical corrections:**

Please proofread for corrections to English grammar and sentence constructions.

Figure 1: The inset pictures are very small, and I think, the ones showing the FOVs are not necessary here, since they are presented in Figure 2. A better figure would be the map and the instrumentation. If the authors do decide to include the pictures of the FOVs, please make sure that the caption states clearly what all the images are. Currently, the caption does not explain what the smaller pictures are.

Generally, figures are rather small and of fairly low resolution because of their inclusion in the pdf. Separately provided figures of a revised document will be bigger in size and of better resolution. However, we tried to improve our figures following the reviewer's comments. Figure 1 now contains a bit more information on the observatories while the FOV/sample images are included in figure 2. The legends were edited accordingly.

Fig. 1. Location of the two study-sites in the Atlantic and the Pacific Ocean, along with some other well-known vent fields for reference purposes. The NEP inset (top) shows the location of the different instrumented nodes of Ocean Networks Canada at the right and the TEMPO-mini ecological module deployed at Main Endeavour Field on the Juan de Fuca Ridge (NEP). The MAR inset (bottom) represents a sketch of the Atlantic observatory (EMSO-Açores) at Lucky Strike vent field on the left and the TEMPO ecological module on the right. For more details of the exact location of the observatories within the hydrothermal vent fields see Matabos et al. (2015) for MAR and Cuvelier et al. (2014) for NEP.

Figure 2: In addition to the sketches with the animals and substrates interpreted, one sample image in its original form, without interpretations drawn in, needs to be included as well for each site. Ideally, instead of a composite sketch, just one sample image should be presented, with and without the interpretations drawn in (and a reference can be made to Table 2 for a comprehensive list of animals seen at the two sites). This provides the opportunity to see what is being analyzed. These images also need scale bars. And, the white arrow that is mentioned in the caption, which is

supposed to be pointing to the fluid exit, is not in the figure. Additionally, there is no mention whatsoever, of 'mussel background' anywhere in the text but it is drawn in in this figure.

Sample images from Figure 1 have now been included in Figure 2 taking into account the comments of both reviewer's 1 and 2. This facilitates interpretation and enhances readability of the scale. Legends have been changed accordingly.